# Unlocking the Potential of Continual Model Merging: An ODE Perspective

**Lihong Lin**[1]   **Haidong Kang**[1][2]

## Abstract

Continual Model Merging (CMM) enables rapid customization of foundation models by sequentially incorporating task-adapted models without repeated retraining. However, existing merging rules usually update the deployed model through fixed algebraic or projection-based operations, providing limited control over how much previously accumulated knowledge should be retained relative to the incoming task model. This limitation leads to unstable retention and performance degradation in long task streams, and becomes more pronounced when tasks have heterogeneous utilities. We propose ODE-driven Merging (ODE-M), a controllable framework that formulates each continual merge as a trajectory in parameter space rather than a one-step endpoint update. Motivated by mode connectivity, ODE-M constructs a barrier-aware trajectory using a rectified time-dependent velocity field, where lightweight first-order feedback from a small calibration set suppresses loss-increasing motion while preserving progress toward the incoming model. The next merged model is then obtained by selecting an operating point along this trajectory through a utility-aware time schedule, providing an explicit mechanism for balancing retained historical knowledge and incoming task expertise. Extensive experiments on standard CMM benchmarks show that ODE-M consistently improves over strong continual merging baselines across CLIP ViT backbones, stream lengths, and heterogeneous task-utility settings.

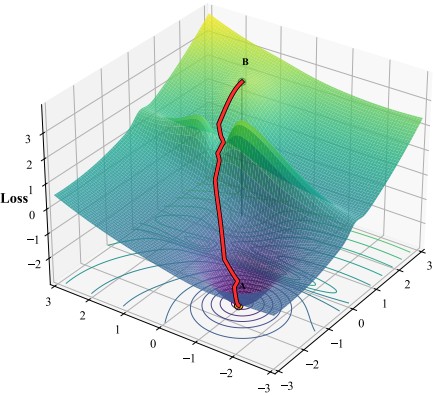

*Figure 1.* **Mode connectivity** suggests that two independently trained models can often be joined by a continuous path in parameter space along which the loss remains low (nearly unchanged).

## 1. Introduction

Fine-tuning has become the standard route for adapting foundation models to various downstream tasks and domains, and the open-source ecosystem has made task-adapted variants increasingly abundant (Houlsby et al., 2019; Wortsman et al., 2022b). As these specialized models accumulate, it becomes desirable to reuse and combine existing task expertise rather than retraining from scratch whenever users' requirements change (Ilharco et al., 2022; Yadav et al., 2023b). Model merging addresses this goal by integrating multiple models with shared architectures directly in parameter space, aiming to consolidate complementary capabilities into a single deployable model without additional training (Li et al., 2023; Yang et al., 2024; Yadav et al., 2024a). Recent studies have demonstrated the utility of model merging across a range of scenarios, from combining language models with complementary task expertise (Fu et al., 2025; Nobari et al., 2025) to consolidating vision models adapted to distinct domains (Ye et al., 2023b; Li et al., 2024).

Conventional model merging typically considers a one-shot setting where multiple task-adapted models are available simultaneously. Among them, previous approaches have studied a broad spectrum of merging rules, ranging from simple weight-space combinations such as weight averag-

---

[1]Northeastern University, Shenyang, China [2]Hebei Key Laboratory of Marine Perception Network and Data Processing, Northeastern University at Qinhuangdao 066004, Hebei Province, China. Correspondence to: Haidong Kang <kang-haidong@qhd.neu.edu.cn>.

*Proceedings of the $43^{rd}$ International Conference on Machine Learning*, Seoul, South Korea. PMLR 306, 2026. Copyright 2026 by the author(s).

Code is available at: https://github.com/yohbii/ODE-M

ing (Wortsman et al., 2022a) and task vector arithmetic (Ilharco et al., 2022) to more structured approaches that improve compatibility via parameter alignment (Ainsworth et al., 2022) or reweighting (Yadav et al., 2023a; Yang et al., 2023). By contrast, continual model merging considers a sequential setting in which task-adapted models arrive over time, and the merged model is updated iteratively after each new task (Li et al., 2025). Recent methods designed for CMM achieve the sequential update either by extending task arithmetic to continually accumulate task vectors relative to a base model, or by introducing projection-based mechanisms that filter interfering components during each merge (Tang et al., 2025b; Yang et al., 2025).

Although these methods have taken the first step toward CMM, the core control problem in continual merging remains underexplored. At each merge step, the deployed model $\Psi_k$ already encodes the knowledge accumulated from previous tasks, while the incoming model $\psi_{k+1}$ brings expertise for a new task (Yang et al., 2024). The update therefore needs to determine not only *where* the next merged model should be, but also *how much* previously accumulated knowledge should be preserved relative to the incoming task. This issue becomes more pronounced in long task streams, where repeated updates gradually amplify small allocation errors, and in heterogeneous-utility scenarios, where degradation on different tasks can have different practical costs, e.g., due to different data scales, user traffic, or reliability requirements (Yadav et al., 2024b).

Existing CMM methods usually specify the update as a fixed rule that maps the current model $\Psi_k$ and the incoming model $\psi_{k+1}$ directly to the next model $\Psi_{k+1}$ (Yang et al., 2024). Such an endpoint-only view provides limited control over two important aspects of the merge. First, it does not explicitly regulate the transition taken between the current and incoming models; if the update passes through high-loss regions in parameter space, the resulting model may suffer from functional degradation and interference with previously merged behaviors (Lee et al., 2025; Hammoud et al., 2024). Second, it provides only a coarse handle on the stability–plasticity trade-off, since the relative contribution of the previous merged model and the incoming task model is implicitly determined by the merging rule rather than by an explicit operating point.

We address this limitation by viewing each continual merge as a controlled transition in parameter space. For a single update, let $\theta_0 = \Psi_k$ denote the current merged model and $\theta_1 = \psi_{k+1}$ denote the incoming task model. Instead of immediately committing to a single endpoint update, we construct a continuous trajectory from $\theta_0$ toward $\theta_1$ and select an operating point on this trajectory. This trajectory perspective exposes two control handles: the shape of the path controls whether the update avoids high-loss regions,

while the operating time controls the relative amount of old and new knowledge incorporated into the next merged model.

Mode connectivity motivates the feasibility of this view. Prior work shows that independently trained solutions can often be connected by continuous curves along which the loss remains low, even when naive linear interpolation exhibits a significant loss barrier (Vrabel et al., 2024; Ren et al., 2025; Tran et al., 2025). This suggests that the difficulty of merging is not solely determined by the endpoints, but also by the choice of the connecting path. However, existing connectivity-based methods typically construct such paths by explicitly parameterizing and optimizing a curve between two endpoints (Khan et al., 2024). While useful for offline analysis, repeatedly solving a separate path-optimization problem is less suitable for online continual merging, where new task models arrive sequentially and the merge procedure must remain lightweight.

To this end, we propose ODE-driven Merging (ODE-M), which generates the connecting path as the trajectory of a time-dependent velocity field. Starting from a base velocity that moves the current model toward the incoming model, ODE-M uses first-order feedback from a small calibration set to locally rectify the trajectory. Specifically, at each integration step, we decompose the velocity into a component aligned with the loss gradient and a component orthogonal to it, and dampen only the loss-increasing component. This produces a barrier-aware trajectory that controls loss escalation while still progressing toward the incoming model. The next merged model is then obtained by selecting an operating time along the trajectory, which naturally implements the desired allocation between retained historical knowledge and incoming task expertise.

To sum up, the main contributions are:

- We formulate CMM from a trajectory perspective and extend the notion of loss barriers from linear interpolation to general connecting paths.

- We propose a novel barrier-aware ODE dynamics via a rectified velocity field, which regulates loss-increasing motion using lightweight first-order feedback.

- We validate our approach with extensive experiments and ablations, showing consistent gains over strong continual merging baselines across CLIP ViT backbones and stream lengths ( as shown in Fig. 2).

## 2. Rethinking the CMM Objective

### 2.1. Problem Formulation

In this paper, we consider a CMM setting in which task-adapted models arrive sequentially over time. Let

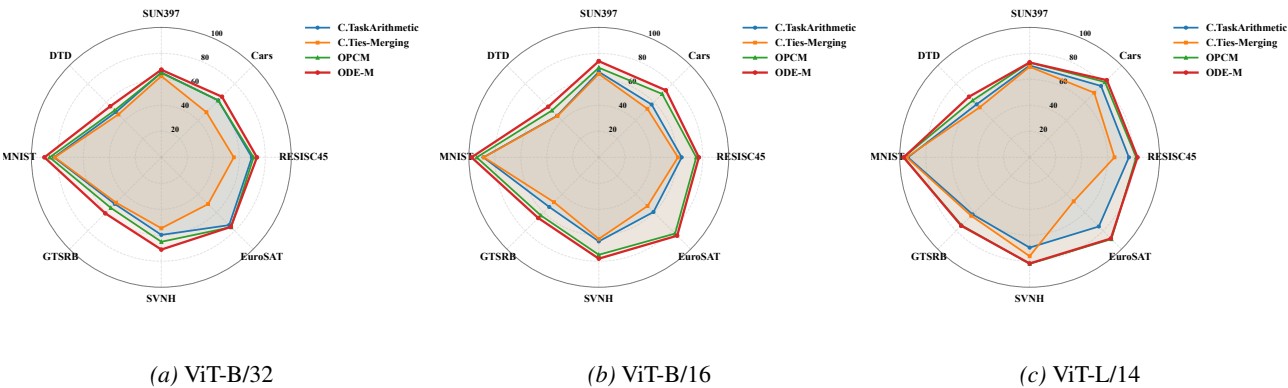

*Figure 2.* **Per-task performance on 8-task continual merging.** Radar plots show the per-task accuracy of different continual merging methods on an 8-task stream for three CLIP ViT backbones: (a) ViT-B/32, (b) ViT-B/16, and (c) ViT-L/14.

$\{\psi_k\}_{k=1}^K \subset \mathbb{R}^d$ denote a stream of trained models, where each $\psi_k$ is specialized for a task or domain $\mathcal{T}$ and all models share the same parameter space. We maintain a deployed merged model $\Psi_k$ after receiving the first $k$ models and update it online when a new model arrives:

$$\Psi_{k+1} = \mathcal{M}(\Psi_k, \psi_{k+1}, \psi_0), \quad \Psi_1 = \psi_1 \quad (1)$$

where $\mathcal{M}$ denotes the merging algorithm, and $\psi_0$ denotes the weights of the shared pre-trained model.

### 2.2. Objective Mismatch in CMM

Existing CMM benchmarks report performance by macro-averaging over the tasks observed so far. This convention is not merely a reporting choice. It implicitly fixes a particular objective. Specifically, macro-average is equivalent to evaluating the deployed model under a *uniform mixture* over tasks, where each task is assumed to be equally likely to occur and equally costly to degrade. While convenient, this is a strong modeling assumption and need not reflect the objective that a continual system ultimately optimizes.

In contrast, a more general and faithful statement is to model system-level performance as an *expected utility* under a non-uniform task mixture. Tasks can contribute unevenly for two simple and widely applicable reasons: **(1)** tasks may appear with different frequencies in the stream or at inference time, so errors on frequent tasks dominate the overall expected objective. **(2)** Even at the same frequency, different tasks can carry different importance, meaning that the same performance drop can have a different impact on the overall objective. Under either interpretation, treating tasks as one vote each can misrank models: two merged models may have the same macro-average yet exhibit substantially different system-level utility.

To make this heterogeneity explicit, we associate each task $\mathcal{T}_i$ with a non-negative weight $w_i$ that represents its relative contribution to the objective. Let $a_{k,i}$ denote the performance of the deployed model $\Psi_k$ on task $\mathcal{T}_i$ after merging the first $k$ tasks. We define the utility-weighted average performance as:

$$\text{ACC}_w(\Psi_k) = \frac{1}{\sum_{i=1}^k w_i} \sum_{i=1}^k w_i \, a_{k,i}, \quad (2)$$

and the utility-weighted backward transfer as

$$\text{BWT}_w(\Psi_k) = \frac{1}{\sum_{i=1}^{k-1} w_i} \sum_{i=1}^{k-1} w_i \, (a_{k,i} - a_{i,i}). \quad (3)$$

The standard macro-average metrics are recovered as a special case when $w_i$ are uniform. This formulation clarifies what controllability should mean in continual merging.

## 3. Continual Model Merging via ODE

### 3.1. Paths and Barriers

Motivated by the ODE perspective, we recast continual model merging as the problem of seeking an optimal continuous path in parameter space (Daheim et al., 2023). Specifically, for a single update step between two models, we denote the currently deployed model $\Psi_k$ by $\theta_0$ and the newly arrived task-adapted model $\psi_{k+1}$ by $\theta_1$.

By contrast, continual model merging poses distinct challenges because tasks arrive sequentially and independently. Unlike the isolated merging of two models, continual merging must incorporate each new model while preserving the capabilities of all previously merged models; otherwise, catastrophic forgetting can occur. However, most existing model merging methods still focus on identifying an optimal fusion strategy for a single pair of models at a time. Consequently, they do not explicitly treat all models as a unified whole and offer limited control over each model's contri-

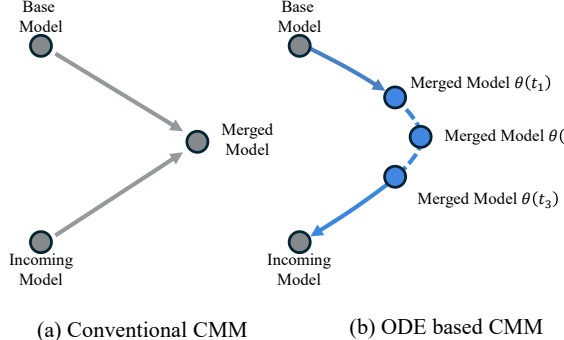

(a) Conventional CMM      (b) ODE based CMM

*Figure 3.* **Conventional vs. ODE-based CMM.** Conventional CMM merges two models in a single step, whereas ODE-based CMM follows a continuous trajectory between endpoints and selects operating points $\theta(t)$ along the path for controllable updates.

bution during the merging process, which can exacerbate forgetting.

In contrast, this paper formulates CMM as the problem of searching for an optimal merging path in parameter space that jointly accounts for all models, as illustrated in Fig. 3. This formulation enables explicit and fine-grained controllability over the merging process and thereby helps alleviate model forgetting. Concretely, we seek to construct a continuous path in parameter space:

$$\theta : [0, 1] \to \mathbb{R}^d, \quad \theta(0) = \theta_0, \theta(1) = \theta_1, \quad (4)$$

With this formulation, each update can be performed in a controlled manner while avoiding high-loss regions during the transition.

**Definition 3.1** (Loss Barrier). Prior work (Entezari et al., 2021) defines the *loss barrier* $B(\theta_0, \theta_1)$ along the linear path between $\theta_0$ and $\theta_1$ as the largest deviation between the loss on the interpolated parameters and the linear interpolation of the endpoint loss:

$$\sup_t [\mathcal{L}(t\theta_1 + (1-t)\theta_0)] - [t\mathcal{L}(\theta_1) + (1-t)\mathcal{L}(\theta_0)]. \quad (5)$$

To enable controllable updates, we do not restrict the transition to linear interpolation. We therefore extend the above notion to an arbitrary continuous connecting path.

**Definition 3.2** (Loss Barrier along Path). Given a continuous path $\theta : [0, 1] \to \mathbb{R}^d$ with $\theta(0) = \theta_0$ and $\theta(1) = \theta_1$, we define the loss barrier of $\theta(\cdot)$ as the maximum excess loss encountered along the path relative to the linear interpolation of the endpoint losses:

$$B(\theta) = \sup_t [\mathcal{L}(\theta(t))] - [t\mathcal{L}(\theta_1) + (1-t)\mathcal{L}(\theta_0)]. \quad (6)$$

With this definition, we seek to construct an update path that keeps the transition stable by minimizing its loss barrier:

$$\min_{\theta(\cdot)} B(\theta) \quad \text{s.t.} \quad \theta(0) = \theta_0, \ \theta(1) = \theta_1. \quad (7)$$

Once such a low-barrier path is available, CMM reduces to selecting an operating point along it, which provides a direct control handle over the update trade-off. To instantiate this principle, we next construct a continuous velocity field whose induced trajectory yields a barrier-aware path.

### 3.2. Constructing the Velocity Field

We start from the simplest case where the connecting path between the two endpoints is the linear interpolation. In this case, the path can be written as:

$$\theta(t) = t\theta_1 + (1-t)\theta_0. \quad (8)$$

This representation immediately induces a constant velocity field along the path:

$$\frac{\mathrm{d}\theta}{\mathrm{d}t} = \theta_1 - \theta_0. \quad (9)$$

We then extend this velocity to the entire parameter space by defining a base vector field:

$$\frac{\partial\theta}{\partial t} = u_t(\theta) = \alpha(t)(\theta_1 - \theta), \quad (10)$$

where $\alpha(t)$ serves as a velocity scheduler, and we set $\alpha(t) = \frac{1}{1-t}$ to ensure that the vector field yields a constant velocity.

However, we find that strictly following the aforementioned linear path ignores the underlying loss landscape $\mathcal{L}(\theta)$, leading to high loss barriers and, in turn, forgetting. To address this issue, we explicitly decouple geometric convergence from loss dynamics. Let $g_t = \nabla_{\theta_t}\mathcal{L}$ denote the loss gradient. We decompose $u_t$ by projecting it onto the subspace spanned by $g_t$. Defining the projection operator $\mathcal{P}_{g_t}(\cdot)$, the decomposition is:

$$u_{\parallel} = \mathcal{P}_{g_t}(u_t) \triangleq \frac{\langle u_t, g_t \rangle}{\|g_t\|^2} g_t, \quad \text{and} \quad u_{\perp} = (I - \mathcal{P}_{g_t})u_t, \quad (11)$$

where $u_{\parallel}$ is the component that changes the loss most directly, whereas $u_{\perp}$ corresponds to transport along the tangent space of the local iso-loss manifold. To reconcile the geometric objective with the loss constraints, we introduce an adaptive factor $\gamma(\theta, t) \in [0, 1]$ to modulate the gradient-aligned component. The resulting dynamics are:

$$v_t(\theta) = u_{\perp} + \gamma(\theta, t)u_{\parallel}. \quad (12)$$

We define the update path as the ODE trajectory induced by this rectified field, i.e., by integrating $\frac{\partial\theta}{\partial t} = v_t(\theta)$ over $t \in [0, 1)$ with $\theta(0) = \theta_0$. Fig. 4 illustrates the induced transport before and after rectification. Specifically, we determine $\gamma(\theta, t)$ by evaluating the alignment between the instantaneous loss gradient $g_t$ and the velocity field $u_t$. The inner product $\langle g_t, u_t \rangle$ is the first-order directional derivative

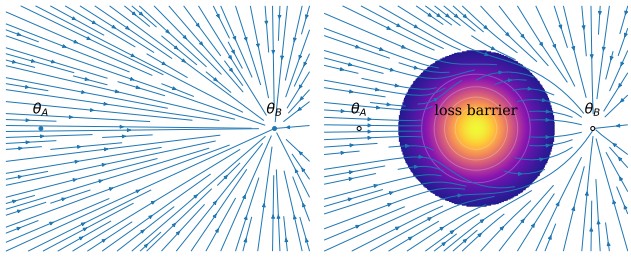

*Figure 4.* **Barrier-aware rectification of the velocity field.** Left: the base field $u_t(\theta)$ induces an unconstrained transport toward the target model $\theta_1$. Right: after gradient projection and adaptive damping, the rectified field $v_t(\theta)$ suppresses loss-increasing components, yielding trajectories that avoid high-loss (barrier) regions while still progressing toward $\theta_1$.

of the loss along the base motion. A positive value indicates that moving toward the target locally increases the loss, while a non-positive value indicates that the base motion is locally loss-decreasing or loss-neutral:

- $\langle g_t, u_t \rangle \leq 0$: The velocity toward $\theta_1$ decreases or maintains the loss. In this regime, the geometric objective aligns with the optimization objective, allowing for unrestricted movement. We have:

$$\gamma(\theta, t) = 1, \text{ if } \langle g_t, u_t \rangle \leq 0. \quad (13)$$

- $\langle g_t, u_t \rangle > 0$: Moving toward the target necessitates climbing the loss surface, contributing to the loss barrier. Here, the velocity component $u_\parallel$ must be damped to ensure the loss increase remains within a prescribed budget. Let $\Delta\mathcal{L} = \mathcal{L}(\theta_1) - \mathcal{L}(\theta_0)$, and we have:

$$\gamma(\theta, t) = \text{clip}\left(\frac{\Delta\mathcal{L}}{\langle g_t, u_t \rangle}, 0, 1\right), \text{ if } \langle g_t, u_t \rangle > 0. \quad (14)$$

Integrating $\frac{\partial\theta}{\partial t} = v_t(\theta)$ from $\theta(0) = \theta_0$ yields a trajectory toward $\theta_1$ whose loss-increasing component is explicitly regulated by $\gamma(\theta, t)$.

### 3.3. Theoretical Analysis

In this section, we justify that the ODE-induced trajectory produced by our rectified velocity field is a valid connecting path for continual model merging. Concretely, we establish two properties. First, the trajectory admits an endpoint extension and converges to the target model $\theta_1$ as $t \to 1$. Second, along this trajectory, the loss barrier (as defined in Section 3.1) is bounded, implying that the transition does not traverse arbitrarily high-loss regions.

**Theorem 3.3** (Convergence of Velocity Field). *Let $\theta(t)$ be the solution of the proposed dynamics $v_t(\theta)$ on $t \in$*

$[0, 1)$ *with $\theta(0) = \theta_0$ and target $\theta_1$. Assume that $g_t = \nabla\mathcal{L}(\theta(t)) \neq 0$ for almost every $t \in [0, 1)$. Moreover, assume there exists a constant $\rho \in [0, 1)$ such that, along the trajectory, for all $t \in [0, 1)$, $\frac{\|\mathcal{P}_{g_t}(u_t)\|}{\|u_t\|} \leq \rho$. Then the trajectory admits a continuous endpoint extension at $t = 1$ and converges to the target model:*

$$\lim_{t \to 1} \theta(t) = \theta_1. \quad (15)$$

**Theorem 3.4** (Bounded Loss Barrier). *Let $\theta(t)$ be the trajectory generated by $\dot{\theta}(t) = v_t(\theta(t))$ on $t \in [0, 1)$ with $\theta(0) = \theta_0$ and a continuous endpoint extension satisfying $\theta(1) = \theta_1$, and assume that $\gamma(\theta, t)$ and $\mathcal{L}$ are differentiable. Then for all $t \in [0, 1)$, we have:*

$$\mathcal{L}(\theta(t)) \leq \mathcal{L}(\theta_0) + t \cdot \max\{\Delta\mathcal{L}, 0\}. \quad (16)$$

*Consequently, the loss barrier along the path satisfies:*

$$B(\theta) \leq \max\{\mathcal{L}(\theta_0) - \mathcal{L}(\theta_1), 0\}. \quad (17)$$

Detailed Proofs are provided in Appendix A.

### 3.4. Time Scheduling along the Trajectory

Determining how far to move along the trajectory is crucial for controlling the stability–plasticity trade-off. A smaller operating time keeps the merged model closer to the previously accumulated model, thereby preserving more historical knowledge, whereas a larger operating time assigns more influence to the incoming task model. Therefore, the operating time should reflect the relative contribution of the incoming task within the current stream.

Let $w_k$ denote the utility weight of the $k$-th task, and let $W_{k-1} = \sum_{i=1}^{k-1} w_i$ denote the accumulated utility of previously merged tasks. Since the current model $\Psi_{k-1}$ summarizes the knowledge from the first $k - 1$ tasks, the incoming model $\psi_k$ should contribute in proportion to its relative utility within the prefix. This leads to the following utility-aware time schedule:

$$t_k = \frac{w_k}{W_{k-1} + w_k} = \frac{w_k}{\sum_{i=1}^{k} w_i}, \quad \Psi_k = \theta(t_k). \quad (18)$$

In the standard equal-utility setting, where $w_1 = \cdots = w_k$, Eq. 18 reduces to $t_k = 1/k$. Thus, as more tasks are merged, the incoming task receives a diminishing operating time, while the retained contribution of the previous merged model increases as $(k - 1)/k$. This schedule turns the integration time into an explicit control handle for allocating contribution between accumulated historical knowledge and incoming task expertise. We empirically validate this behavior in Appendix B by showing that the retained contribution implied by the empirically selected operating times strongly correlates with the equal-utility retention ratio $(k - 1)/k$.

# 4. Experiments

In this section, we conduct extensive experiments in the continual model merging setting. We further perform ablation studies to analyze the behavior of our method.

## 4.1. Experimental Setup

**Models and Datasets**. Following OPCM (Tang et al., 2025b), we consider CLIP vision backbones (Radford et al., 2021) with three architectures: ViT-B/32, ViT-B/16, and ViT-L/14. We obtain the stream of task-adapted models and the corresponding datasets from FusionBench (Tang et al., 2025a) by fine-tuning each pre-trained backbone sequentially on 20 tasks, and evaluate continual model merging on task streams of length 8, 14, and 20. Additional experimental details are provided in Appendix B.

**Evaluation Metrics**. We evaluate continual model merging using two standard metrics: average accuracy (ACC) and backward transfer (BWT) (Lin et al., 2022), which quantify overall performance and retention of earlier tasks after sequential updates.

**Baselines**. We compare against four representative continual merging baselines, including parameter Averaging (SWA), Continual Task Arithmetic (Ilharco et al., 2022), Continual Ties-Merging (Yadav et al., 2023b), and OPCM (Tang et al., 2025b).

**Implementation Details**. We maintain a small calibration dataset that aggregates samples from previously seen tasks, with a total of 1024 examples. We integrate the proposed ODE using the Euler method with a fixed step size of 0.05.

## 4.2. CMM Performance on CLIP ViT Backbones

We evaluate ODE-M on CLIP vision backbones, including ViT-B/32, ViT-B/16, and ViT-L/14. For each backbone, we construct a set of task-specific fine-tuned checkpoints starting from the same CLIP pre-trained weights and evaluate continual merging on task-stream prefixes of length 8, 14, and 20. To account for stochasticity induced by task ordering, we repeat each setting over 10 randomly permuted task sequences and report aggregated performance.

Table 1 reports the performance of different continual merging rules under varying stream lengths and backbones, measured by average accuracy and backward transfer. Across all three architectures, ODE-M achieves the highest overall accuracy and ranks first for every stream length. For example, on ViT-B/32, ODE-M improves ACC from 75.5 to 79.6 on 8 tasks, from 71.9 to 74.1 on 14 tasks, and from 65.7 to 67.1 on 20 tasks, compared to the strongest baseline OPCM. Similar gains hold on ViT-B/16, where ODE-M reaches 85.3, 78.5, and 70.4 ACC for 8/14/20 tasks, and on ViT-L/14, where it attains 87.7, 85.8, and 81.1 ACC, indicating

that the advantage persists as model capacity increases.

In terms of forgetting, all methods exhibit negative BWT as the stream grows, and ODE-M remains competitive with or better than prior baselines in most settings, especially for shorter streams. While improving ACC can come with a more aggressive update and thus more negative BWT in some long-stream configurations, the overall results suggest that our method provides a favorable accuracy–stability trade-off across architectures and task horizons.

## 4.3. CMM under Heterogeneous Task Utility

To study continual model merging beyond the macro-averaged convention, we introduce heterogeneous task utility by assigning each task $\mathcal{T}_i$ a non-negative importance weight $w_i$. In each run, we randomly sample $\{w_i\}$ from a fixed distribution and normalize them so that $\sum_i w_i = 1$, then evaluate continual merging under the resulting non-uniform task mixture using the weighted metrics defined in Section 2.2. We repeat this procedure over 10 independent draws of $\{w_i\}$. For a fair comparison, we adjust all baselines to optimize the same utility-weighted objective. In particular, we modify their sequential combination coefficients to respect the task weights $\{w_i\}$, ensuring that each method is evaluated under an identical notion of task utility rather than benefiting from a mismatched uniform objective. Implementation details of these utility-aware baseline variants are provided in Appendix D.

Table 2 summarizes the utility-weighted results. Across all backbones and stream lengths, ODE-M consistently achieves the highest $\mathrm{ACC}_w$, indicating that it better optimizes the non-uniform task mixture induced by $\{w_i\}$. In particular, compared to the strongest baseline OPCM, ODE-M improves $\mathrm{ACC}_w$ from 74.2 to 77.9 on ViT-B/32 with 8 tasks, from 81.0 to 84.6 on ViT-B/16 with 8 tasks, and yields a pronounced gain on the longest ViT-L/14 stream, boosting $\mathrm{ACC}_w$ from 75.0 to 80.0 at 20 tasks. Beyond average utility, ODE-M also maintains competitive retention under heterogeneous utility. While some baselines can attain less negative $\mathrm{BWT}_w$ in certain configurations, they do so at a clear cost in $\mathrm{ACC}_w$ under the same weighted objective. By contrast, ODE-M preserves a favorable utility-stability trade-off, and improves upon OPCM in $\mathrm{BWT}_w$ across all settings (e.g., from $-5.9$ to $-4.8$ on ViT-B/32 with 8 tasks and from $-5.9$ to $-4.8$ on ViT-L/14 with 20 tasks).

## 4.4. Ablation Study & Efficiency Analysis

We conduct ablation studies to validate the robustness of our method to key practical factors. In particular, we vary the number of calibration samples and the numerical integration accuracy to assess the sensitivity of performance to calibration budget and solver fidelity. In addition, to validate the efficiency of our method, we conduct an efficiency analysis.

*Table 1.* Comparison of continual merging performance on CLIP ViT backbones. We report the final macro-averaged accuracy (ACC) and backward transfer (BWT) after merging task streams of length 8, 14, and 20 on ViT-B/32, ViT-B/16, and ViT-L/14. The prefix "C." indicates continual variants.

| | Methods | ViT-B/32 | | | ViT-B/16 | | | ViT-L/14 | | |
|---|---|---|---|---|---|---|---|---|---|---|
| | | 8 tasks | 14 tasks | 20 tasks | 8 tasks | 14 tasks | 20 tasks | 8 tasks | 14 tasks | 20 tasks |
| | Pre-Trained | 48.1 | 56.9 | 55.6 | 55.4 | 62.0 | 59.8 | 64.9 | 69.1 | 65.6 |
| | Fine-Tuned | 90.4 | 89.3 | 89.8 | 92.4 | 91.3 | 91.6 | 94.3 | 93.4 | 93.5 |
| | C. Fine-Tuned | 79.8 | 67.4 | 62.6 | 82.9 | 72.2 | 68.2 | 90.0 | 70.9 | 77.7 |
| ACC↑ | Average (SWA) | 66.3±0.0 | 65.4±0.0 | 61.1±0.0 | 72.3±0.0 | 69.7±0.0 | 64.8±0.0 | 80.0±0.0 | 77.5±0.0 | 71.1±0.0 |
| | C. Task Arithmetic | 67.5±0.0 | 66.5±0.0 | 60.6±0.0 | 77.1±0.0 | 70.9±0.0 | 64.2±0.0 | 82.1±0.0 | 77.9±0.0 | 70.3±0.0 |
| | C. Ties-Merging | 49.0±10.2 | 66.2±0.6 | 59.9±0.7 | 66.8±3.7 | 70.5±0.8 | 63.0±1.6 | 64.3±7.0 | 78.0±0.6 | 68.3±0.9 |
| | OPCM | 75.5±0.5 | 71.9±0.3 | 65.7±0.2 | 81.8±0.3 | 77.1±0.5 | 70.3±0.2 | 87.0±0.4 | 83.5±0.2 | 76.0±0.2 |
| | **ODE-M(Ours)** | **79.6**±1.9 | **74.1**±1.5 | **67.1**±0.8 | **85.3**±0.6 | **78.5**±0.9 | **70.4**±0.5 | **87.7**±0.9 | **85.8**±1.2 | **81.1**±1.0 |
| BWT↑ | Average (SWA) | -11.5±2.2 | -8.0±1.3 | -7.1±2.1 | -9.7±1.5 | -7.1±1.4 | -7.3±1.7 | -7.3±1.4 | -5.8±1.0 | -6.4±1.5 |
| | C. Task Arithmetic | -9.6±1.5 | -1.3±0.3 | -3.4±0.4 | -4.2±1.0 | -1.3±0.4 | -3.6±0.4 | -7.1±0.8 | -1.8±0.3 | -3.3±0.3 |
| | C. Ties-Merging | -15.3±8.0 | 1.9±0.6 | -1.5±0.7 | -5.5±0.4 | 1.4±0.7 | -1.5±1.2 | -13.0±5.7 | 1.1±0.4 | -2.9±1.0 |
| | OPCM | -6.3±1.1 | -6.0±1.0 | -7.8±1.5 | -4.8±0.7 | -5.1±1.4 | -6.3±3.8 | -2.6±1.0 | -4.3±0.7 | -6.5±1.8 |
| | **ODE-M(Ours)** | -4.2±3.9 | -7.8±3.2 | -9.4±2.2 | -6.6±2.8 | -8.1±3.0 | -9.9±3.5 | -4.0±2.4 | -7.4±2.5 | -10.3±2.7 |

*Table 2.* Utility-weighted continual merging results under heterogeneous task utility. We report weighted average accuracy ($\text{ACC}_w$) and weighted backward transfer ($\text{BWT}_w$) after 8-, 14-, and 20-task streams on CLIP ViT backbones. The prefix "C." indicates continual variants.

| | Methods | ViT-B/32 | | | ViT-B/16 | | | ViT-L/14 | | |
|---|---|---|---|---|---|---|---|---|---|---|
| | | 8 tasks | 14 tasks | 20 tasks | 8 tasks | 14 tasks | 20 tasks | 8 tasks | 14 tasks | 20 tasks |
| $\text{ACC}_w$ ↑ | Average (SWA) | 65.1±0.9 | 64.3±0.8 | 60.2±0.7 | 71.5±0.9 | 69.2±0.8 | 63.7±0.8 | 79.2±0.8 | 76.8±0.7 | 70.0±0.7 |
| | C. Task Arithmetic | 66.8±0.8 | 65.7±0.9 | 60.0±0.8 | 76.3±0.8 | 70.5±0.7 | 63.4±0.8 | 81.5±0.7 | 77.1±0.7 | 69.5±0.7 |
| | C. Ties-Merging | 58.4±6.1 | 65.0±1.4 | 59.1±1.2 | 69.1±3.0 | 70.0±1.2 | 62.0±1.5 | 72.0±4.5 | 77.3±1.0 | 67.0±1.2 |
| | OPCM | 74.2±0.7 | 70.8±0.6 | 64.5±0.5 | 81.0±0.6 | 76.5±0.6 | 69.4±0.6 | 86.2±0.6 | 82.9±0.6 | 75.0±0.6 |
| | **ODE-M(Ours)** | **77.9**±1.4 | **73.2**±1.1 | **66.0**±0.9 | **84.6**±0.7 | **78.0**±0.8 | **70.0**±0.6 | **87.0**±0.8 | **84.6**±0.9 | **80.0**±0.8 |
| $\text{BWT}_w$ ↑ | Average (SWA) | -10.8±1.8 | -7.6±1.3 | -6.9±1.6 | -9.0±1.6 | -6.8±1.2 | -7.1±1.4 | -6.8±1.2 | -5.4±0.9 | -6.1±1.2 |
| | C. Task Arithmetic | -8.7±1.5 | -1.6±0.5 | -3.1±0.7 | -3.9±1.2 | -1.5±0.6 | -3.3±0.6 | -5.5±0.8 | -1.9±0.5 | -3.0±0.5 |
| | C. Ties-Merging | -12.0±5.5 | 1.1±0.9 | -1.2±1.0 | -5.0±1.8 | 0.8±0.9 | -1.3±1.3 | -9.5±3.8 | 0.6±0.7 | -2.5±1.1 |
| | OPCM | -5.9±1.0 | -5.6±0.9 | -7.0±1.4 | -4.2±0.9 | -4.6±1.2 | -5.8±2.5 | -2.8±0.8 | -4.1±0.8 | -5.9±1.5 |
| | **ODE-M(Ours)** | -5.8±1.6 | -4.9±1.4 | -8.1±1.7 | -4.5±1.1 | -5.9±1.2 | -6.2±1.9 | -2.5±1.0 | -4.4±1.1 | -7.8±1.4 |

**Sensitivity to Calibration Set Size**. We evaluate the robustness of ODE-M to the calibration budget by varying the number of calibration samples $n$ over a wide range on an 8-task CMM stream. We repeat each setting five times and report the mean accuracy with standard deviation. As shown in Table 3, performance improves steadily as $n$ increases, with diminishing returns beyond a moderate budget. For ViT-B/32, accuracy rises from 62.8 at $n=8$ to 79.4 at $n=512$ and only changes marginally thereafter (79.6 at $n=1024$ and 79.8 at $n=2048$). A similar saturation trend holds for larger backbones, reaching 85.1–85.3 for ViT-B/16 and 87.5–87.8 for ViT-L/14 once $n \geq 512$. These results suggest that ODE-M remains effective under limited calibration data while benefiting from additional samples when available, and that a few hundred samples already provide strong accuracy.

**Sensitivity to Numerical Integration Accuracy**. We study the sensitivity of ODE-M to numerical integration accuracy by varying the Euler step size on CLIP ViT-B/32 with an 8-task stream. For each step size, we run five trials and report mean accuracy with standard deviation. As shown in Table 4, performance is stable across a wide range of small to moderate step sizes. In particular, the accuracy remains around 80% for step sizes from $10^{-3}$ to $10^{-2}$ (e.g., $80.2 \pm 0.5$ at $10^{-3}$ and $80.0 \pm 0.5$ at $10^{-2}$), and degrades only mildly at $5 \times 10^{-2}$ ($79.4 \pm 0.6$). When the discretization becomes overly coarse, accuracy drops more noticeably and variability increases, as observed at $2 \times 10^{-1}$ ($78.7 \pm 0.7$). These results indicate that a reasonably fine step size is sufficient to obtain reliable performance while avoiding unnecessary computation, and we use $5e-2$ in our main experiments as a robust default.

**Efficiency Analysis**. Table 5 validates the efficiency by reporting the average wall-clock time required to merge

*Table 3.* **Sensitivity to calibration set size.** Average accuracy (%) of ODE-M on CLIP ViT backbones under an 8-task continual model merging stream, as a function of the number of calibration samples $n$. Results are reported as mean $\pm$ std over 5 runs.

| Model | Number of Samples | | | | | | | | |
|---|---|---|---|---|---|---|---|---|---|
| | 8 | 16 | 32 | 64 | 128 | 256 | 512 | 1024 | 2048 |
| ViT-B/32 | 62.8±3.5 | 64.9±3.0 | 70.1±2.6 | 73.0±2.3 | 75.0±2.1 | 76.8±2.0 | 79.4±1.9 | 79.6±1.5 | 79.8±1.8 |
| ViT-B/16 | 68.5±1.6 | 70.6±1.3 | 75.8±1.1 | 78.7±0.9 | 80.7±0.8 | 82.5±0.7 | 85.1±0.6 | 85.3±0.4 | 85.1±0.6 |
| ViT-L/14 | 70.9±1.8 | 73.0±1.5 | 78.2±1.3 | 81.1±1.1 | 83.1±1.0 | 84.9±0.9 | 87.5±0.8 | 87.7±0.9 | 87.8±0.9 |

*Table 4.* **Sensitivity to numerical integration step size.** Average accuracy (%) of ODE-M on CLIP ViT-B/32 under an 8-task continual model merging stream, using Euler integration with different step sizes.

| Step size | ViT-B/32 |
|---|---|
| 1e-3 | $80.2 \pm 0.5$ |
| 2.5e-3 | $80.1 \pm 0.5$ |
| 5e-3 | $80.0 \pm 0.5$ |
| 1e-2 | $79.8 \pm 0.5$ |
| 2.5e-2 | $79.6 \pm 0.6$ |
| 5e-2 | $79.4 \pm 0.6$ |
| 1e-1 | $79.5 \pm 0.6$ |
| 2e-1 | $78.7 \pm 0.7$ |

*Table 5.* Runtime overhead of ODE-M across architectures.

| | ViT-B/32 | ViT-B/16 | ViT-L/14 |
|---|---|---|---|
| Overhead (s) | 123.9 | 200.8 | 246.8 |

one incoming task-adapted model into the current merged model over a 20-task stream for each CLIP ViT backbone. As shown, the overhead increases with model size, taking 123.9s, 200.8s, and 246.8s per merge for ViT-B/32, ViT-B/16, and ViT-L/14, respectively. This cost is mainly incurred by the numerical integration procedure together with the per-step gradient computations on the calibration set, and remains practical in our setting where merging is performed only when a new task model arrives.

## 5. Related Works

### 5.1. Continual Model Merging

Model merging has emerged as a scalable paradigm for combining the capabilities of multiple task-specific models into a single model without access to the original training data. Early research primarily focused on the "one-shot" (offline) setting, where all expert models are available simultaneously. Foundational techniques in this domain range from simple weight averaging (Wortsman et al., 2022a) to more structured arithmetic operations such as Task Arithmetic (Ilharco et al., 2022) and interference-resolving methods such as Ties-Merging (Yadav et al., 2023a).

Continual Model Merging (CMM) extends this paradigm to sequential scenarios where task-adapted models arrive in a stream, requiring the deployed model to be updated iteratively (Tang et al., 2025b). Initial approaches to CMM largely adapted static merging rules, such as moving averages or cumulative task-vector addition, to the sequential setting. However, these naive extensions often suffer from catastrophic forgetting due to parameter interference. To address this issue, recent works have introduced projection-based mechanisms. For instance, OPCM (Tang et al., 2025b) employs orthogonal projections to filter conflicting gradient directions, while Yang et al. (2025) proposes dual projections to balance stability and plasticity without data replay. Despite these improvements, existing CMM methods predominantly rely on fixed algebraic combinations that treat model updates as instantaneous jumps in parameter space.

### 5.2. Mode Connectivity and Loss Landscapes

The geometry of the loss landscape plays a pivotal role in understanding the relationship between independently trained neural networks. The theory of mode connectivity posits that isolated local optima found by stochastic gradient descent are not disconnected islands but can often be joined by continuous paths along which the loss remains low (Garipov et al., 2018; Jaiswal et al., 2024). Naive linear interpolation between two models, however, typically traverses a high-loss region referred to as a loss barrier (Entezari et al., 2021). Recent studies demonstrate that non-linear connecting paths can effectively circumvent these barriers. Ren et al. (2025) and Tran et al. (2025) explore such connectivity in complex architectures, showing that barrier-free paths exist even for models with distinct initialization or training trajectories.

Existing approaches exploit these insights to improve model fusion, for instance by aligning parameters to account for permutation symmetries (Ainsworth et al., 2022) or by explicitly optimizing a parametric curve between endpoints (Khan et al., 2024). However, these methods typically treat path finding as a static, offline optimization problem, which is computationally prohibitive in continual settings. Moreover, prior work on connectivity (Vrabel et al., 2024) primarily focuses on existence results or one-shot merging scenarios. Neural ODEs have also been used in multi-task learning, e.g. Ye et al. (2023a) learn task-specific positions in a time-aware ODE flow to mitigate task competition dur-

ing joint training. Different from this training-based multi-task formulation, our work studies training-free continual model merging and uses an ODE-induced trajectory to regulate the parameter-space transition between the current merged model and each incoming task model. In this way, ODE-M makes connectivity-based, barrier-aware updates practical in continual model merging.

## 6. Conclusion and Future Work

In this paper, we propose ODE-driven continual model merging (ODE-M), a controllable framework for merging sequentially arriving task-adapted models. Instead of treating each merge as a fixed algebraic update between isolated checkpoints, ODE-M formulates the update as a continuous trajectory in parameter space and uses lightweight first-order feedback to suppress loss-increasing motion along the path. By constructing barrier-aware trajectories and selecting operating points through a utility-aware time schedule, ODE-M provides an explicit mechanism for balancing retained historical knowledge and incoming task expertise. Extensive experiments on standard continual model merging benchmarks demonstrate consistent improvements over strong baselines across different CLIP ViT backbones and stream lengths. Future work will extend this trajectory-based perspective to larger language and multimodal models, develop more efficient integration strategies to reduce merge-time overhead, and study more practical calibration regimes where only proxy, partial, or dynamically updated calibration data is available.

## Impact Statement

This paper studies continual model merging and proposes an ODE-driven merging procedure that produces a controlled transition between task-adapted models without additional training. The primary positive impact is that such a mechanism can reduce the need for repeated fine-tuning or re-training when new task models arrive, potentially lowering the engineering and computational overhead of maintaining multi-task systems, and making model customization more accessible in resource-constrained settings.

## Acknowledgments

This work is supported by the Fundamental Research Funds for the Central Universities (N2623009).

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

## Appendix Overview

This appendix is organized as follows:

- **Theorem proofs** (Appendix A): full proofs of the convergence and bounded-barrier results.

- **Detailed experiments** (Appendix B): task suites, evaluation protocol, and additional experimental results/visualizations.

- **ODE-based merging procedure** (Appendix C): a step-by-step description and pseudocode of ODE-M.

- **Baseline details under heterogeneous task utility** (Appendix D): how we adapt each baseline to optimize the same utility-weighted objective.

- **Discussions and limitations** (Appendix E): practical considerations, assumptions, and remaining limitations.

## A. Theorem Proofs

**Theorem A.1** (Convergence of Velocity Field). *Let $\theta(t)$ be the solution of the proposed dynamics $v_t(\theta)$ on $t \in [0, 1)$ with $\theta(0) = \theta_0$ and target $\theta_1$. Assume that $g_t = \nabla \mathcal{L}(\theta(t)) \neq 0$ for almost every $t \in [0, 1)$. Moreover, assume there exists a constant $\rho \in [0, 1)$ such that, along the trajectory, for all $t \in [0, 1)$, $\frac{\|\mathcal{P}_{g_t}(u_t)\|}{\|u_t\|} \leq \rho$. Then the trajectory admits a continuous endpoint extension at $t = 1$ and converges to the target model:*

$$\lim_{t \to 1} \theta(t) = \theta_1. \tag{19}$$

*Proof.* Define the error $e(t) \triangleq \theta(t) - \theta_1$. By construction of the base field $u_t(\theta) = \alpha(t)(\theta_1 - \theta)$, we have along the path:

$$u_t(\theta(t)) = -\alpha(t)e(t), \qquad \alpha(t) = \frac{1}{1-t} \tag{20}$$

Recall that the proposed velocity takes the form:

$$v_t(\theta) = u_\perp + \gamma_t u_\| = u_t(\theta) - (1 - \gamma_t)u_\|, \tag{21}$$

where $\gamma_t \in [0, 1]$ and $u_\| = \mathcal{P}_{g_t}(u_t)$ with $g_t = \nabla \mathcal{L}(\theta(t))$. Therefore, the error dynamics satisfy $\dot{e}(t) = \dot{\theta}(t) = v_t(\theta(t))$, and hence

$$\dot{e}(t) = u_t(\theta(t)) - (1 - \gamma_t)\mathcal{P}_{g_t}(u_t(\theta(t))). \tag{22}$$

We study the evolution of the squared error $V(t) \triangleq \|e(t)\|^2$. Differentiating and using $\dot{V}(t) = 2\langle e(t), \dot{e}(t) \rangle$ yields:

$$\dot{V}(t) = 2\langle e(t), u_t(\theta(t)) \rangle - 2(1 - \gamma_t)\langle e(t), \mathcal{P}_{g_t}(u_t(\theta(t))) \rangle. \tag{23}$$

The first term is explicitly contractive:

$$2\langle e(t), u_t(\theta(t)) \rangle = 2\langle e(t), -\alpha(t)e(t) \rangle. \tag{24}$$

For the second term, since $\gamma_t \in [0, 1]$, we have $0 \leq 1 - \gamma_t \leq 1$. Applying Cauchy-Schwarz inequality gives:

$$-2(1 - \gamma_t)\langle e(t), \mathcal{P}_{g_t}(u_t) \rangle \leq 2(1 - \gamma_t)\|e(t)\| \, \|\mathcal{P}_{g_t}(u_t)\| \tag{25}$$
$$\leq 2\|e(t)\| \, \|\mathcal{P}_{g_t}(u_t)\|$$

By the given assumption, along the path we have $\|\mathcal{P}_{g_t}(u_t)\| \leq \rho \|u_t\|$ for $\rho \in [0, 1)$. Moreover, $\|u_t(\theta(t))\| = \alpha(t)\|\theta_1 - \theta(t)\| = \alpha(t)\|e(t)\|$. Substituting these into Eq.(25) yields:

$$-2(1 - \gamma_t)\langle e(t), \mathcal{P}_{g_t}(u_t) \rangle \leq 2\rho \, \alpha(t)\|e(t)\|^2 = 2\rho \, \alpha(t)V(t). \tag{26}$$

Combining Eq.(23), Eq.(24), and Eq.(26), we obtain the differential inequality:

$$\dot{V}(t) \le -2(1-\rho)\alpha(t)V(t) = -\frac{2(1-\rho)}{1-t}V(t). \tag{27}$$

Let $W(t) \triangleq \log V(t)$ on the set where $V(t) > 0$ (the case $V(t) \equiv 0$ is trivial). Dividing both sides of Eq.(27) by $V(t)$ gives:

$$\dot{W}(t) \le -\frac{2(1-\rho)}{1-t}. \tag{28}$$

Integrating from 0 to $t \in [0, 1)$ yields:

$$\log V(t) - \log V(0) \le -2(1-\rho)\int_0^t \frac{1}{1-s}ds = 2(1-\rho)\log(1-t). \tag{29}$$

Exponentiating both sides gives:

$$V(t) \le V(0)\,(1-t)^{2(1-\rho)}. \tag{30}$$

Taking square roots, we obtain the claimed quantitative bound:

$$\|e(t)\| \le \|e(0)\|\,(1-t)^{1-\rho}, \quad \forall t \in [0, 1). \tag{31}$$

Since $\rho < 1$, the right-hand side vanishes as $t \to 1$, implying $\|e(t)\| \to 0$ and therefore $\theta(t) \to \theta_1$. This also shows that the path admits a continuous endpoint extension by defining $\theta(1) \triangleq \theta_1$. $\qquad\square$

**Theorem A.2** (Bounded Loss Barrier). *Let $\theta(t)$ be the trajectory generated by $\dot{\theta}(t) = v_t(\theta(t))$ on $t \in [0, 1)$ with $\theta(0) = \theta_0$ and a continuous endpoint extension satisfying $\theta(1) = \theta_1$, and assume that $\gamma(\theta, t)$ and $\mathcal{L}$ are differentiable. Then for all $t \in [0, 1)$, we have:*

$$\mathcal{L}(\theta(t)) \le \mathcal{L}(\theta_0) + t \cdot \max\{\Delta\mathcal{L}, 0\}. \tag{32}$$

*Consequently, the loss barrier along the path satisfies:*

$$B(\theta) \le \max\{\mathcal{L}(\theta_0) - \mathcal{L}(\theta_1), 0\}. \tag{33}$$

*Proof.* By the chain rule, the loss evolution along the trajectory satisfies:

$$\frac{\mathrm{d}}{\mathrm{d}t}\mathcal{L}(\theta(t)) = \left\langle \nabla\mathcal{L}(\theta(t)), \dot{\theta}(t) \right\rangle = \langle g_t, v_t \rangle. \tag{34}$$

Recall that $v_t = u_\perp + \gamma_t u_\|$, where $u_\| = \mathcal{P}_{g_t}(u_t)$ and $u_\perp = (I - \mathcal{P}_{g_t})u_t$ is orthogonal to $g_t$ by construction. Hence $\langle g_t, u_\perp \rangle = 0$, $\langle g_t, u_\| \rangle = \langle g_t, u_t \rangle$, which implies:

$$\frac{\mathrm{d}}{\mathrm{d}t}\mathcal{L}(\theta(t)) = \gamma_t \langle g_t, u_t \rangle. \tag{35}$$

We upper bound the right-hand side by considering the two regimes in the definition of $\gamma(\theta, t)$.

**Case 1:** $\langle g_t, u_t \rangle \le 0$. By definition, $\gamma_t = 1$, so Eq.(35) yields:

$$\frac{\mathrm{d}}{\mathrm{d}t}\mathcal{L}(\theta(t)) = \langle g_t, u_t \rangle \le 0 \le \max\{\Delta\mathcal{L}, 0\}. \tag{36}$$

**Case 2:** $\langle g_t, u_t \rangle > 0$. In this regime, $\gamma_t = \mathrm{clip}\left(\frac{\Delta\mathcal{L}}{\langle g_t, u_t \rangle}, 0, 1\right)$. If $\Delta\mathcal{L} \le 0$, then $\frac{\Delta\mathcal{L}}{\langle g_t, u_t \rangle} < 0$ and clipping gives $\gamma_t = 0$, so $\frac{\mathrm{d}}{\mathrm{d}t}\mathcal{L}(\theta(t)) = 0 \le \max\{\Delta\mathcal{L}, 0\}$. If $\Delta\mathcal{L} > 0$, then $\gamma_t \le \frac{\Delta\mathcal{L}}{\langle g_t, u_t \rangle}$, and thus:

$$\frac{\mathrm{d}}{\mathrm{d}t}\mathcal{L}(\theta(t)) = \gamma_t \langle g_t, u_t \rangle \le \Delta\mathcal{L} = \max\{\Delta\mathcal{L}, 0\}. \tag{37}$$

Therefore, in all cases we conclude that for almost every $t \in [0, 1]$:

$$\frac{\mathrm{d}}{\mathrm{d}t}\mathcal{L}(\theta(t)) \leq \max\{\Delta\mathcal{L}, 0\} \tag{38}$$

Integrating Eq.(38) from 0 to $t$ yields:

$$\mathcal{L}(\theta(t)) - \mathcal{L}(\theta_0) \leq \int_0^t \max\{\Delta\mathcal{L}, 0\}ds = t \cdot \max\{\Delta\mathcal{L}, 0\}, \tag{39}$$

which proves the claimed envelope $\mathcal{L}(\theta(t)) \leq \mathcal{L}(\theta_0) + t \cdot \max\{\Delta\mathcal{L}, 0\}$.

It remains to bound the barrier. By Definition 3.2, we have:

$$B(\theta) = \sup_{t \in [0,1]} \left( \mathcal{L}(\theta(t)) - [(1-t)\mathcal{L}(\theta_0) + t\mathcal{L}(\theta_1)] \right). \tag{40}$$

Using the envelope above and $\Delta\mathcal{L} = \mathcal{L}(\theta_1) - \mathcal{L}(\theta_0)$, we obtain for any $t \in [0, 1)$:

$$\begin{aligned}
\mathcal{L}(\theta(t)) - [(1-t)\mathcal{L}(\theta_0) + t\mathcal{L}(\theta_1)] &\leq \mathcal{L}(\theta_0) + t\max\{\Delta\mathcal{L}, 0\} - \mathcal{L}(\theta_0) - t\Delta\mathcal{L} \\
&= t(\max\{\Delta\mathcal{L}, 0\} - \Delta\mathcal{L}) = t \cdot (-\min\{\Delta\mathcal{L}, 0\}) \\
&\leq -\min\{\Delta\mathcal{L}, 0\} = \max\{\mathcal{L}(\theta_0) - \mathcal{L}(\theta_1), 0\}.
\end{aligned} \tag{41}$$

Taking the supremum over $t \in [0, 1]$ gives:

$$B(\theta) \leq \max\{\mathcal{L}(\theta_0) - \mathcal{L}(\theta_1), 0\}, \tag{42}$$

which concludes the proof. $\square$

# B. Detailed Experiments

### B.1. Experimental Settings

**Task suites**. We evaluate continual model merging on three task suites with increasing lengths. The **8-task** suite contains SUN397 (Xiao et al., 2010), STANFORD-CARS (Krause et al., 2013), RESISC45 (Cheng et al., 2017), EUROSAT (Helber et al., 2019), SVHN (Netzer et al., 2011), GTSRB (Stallkamp et al., 2011), MNIST (LeCun et al., 2002), and DTD (Cimpoi et al., 2014). The **14-task** suite augments the 8-task suite with FLOWERS102 (Nilsback & Zisserman, 2008), PCAM (Veeling et al., 2018), FER2013 (Goodfellow et al., 2013), OXFORD-IIITPET (Parkhi et al., 2012), STL10 (Coates et al., 2011), and CIFAR100 (Krizhevsky et al., 2009). The **20-task** suite further augments the 14-task suite with CIFAR10 (Krizhevsky et al., 2009), FOOD101 (Bossard et al., 2014), FASHIONMNIST (Xiao et al., 2017), EMNIST (Cohen et al., 2017), KMNIST (Clanuwat et al., 2018), and RENDEREDSST2 (Radford et al., 2021). Each task corresponds to a supervised classification dataset. At step $k$, the deployed merged model is evaluated on the test split of all tasks seen so far.

**Backbones and task-adapted models.** Following FusionBench (Tang et al., 2025a), we adopt CLIP image encoders as the shared architecture for all tasks. We report results on `openai/clip-vit-base-patch32` (ViT-B/32), `openai/clip-vit-base-patch16` (ViT-B/16), and `openai/clip-vit-large-patch14` (ViT-L/14) (Radford et al., 2021). Let $\psi_0$ denote the pretrained CLIP vision backbone. For each task $\mathcal{T}_k$, we obtain a task-adapted model $\psi_k$ by fine-tuning the backbone on $\mathcal{T}_k$ with a task-specific classification head. All task models share the same parameter space (same backbone architecture), which enables training-free merging in a continual setting.

**Continual protocol**. Task-adapted models arrive sequentially. After receiving $\psi_k$, the merging algorithm updates the deployed model $\Psi_k$ without additional training on past-task data (unless explicitly stated). We follow the same task suite definitions above for the 8/14/20-task settings.

### B.2. More Experiment Results

**Per-task Accuracy on Longer Streams**. We provide additional per-task accuracy results for longer task streams. The complete per-task accuracies on the 20-task setting across all CLIP ViT backbones are shown in Fig. 7.

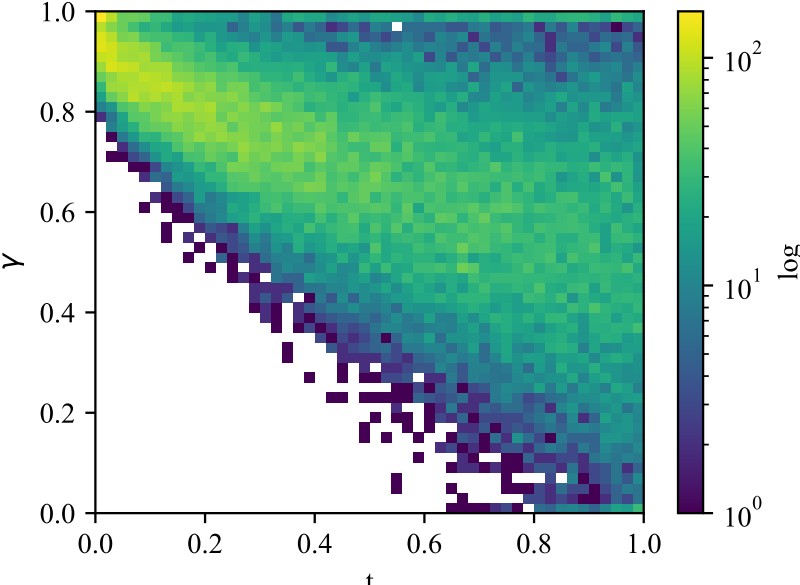

*Figure 5.* Empirical statistics of the adaptive rectification coefficient $\gamma(\theta, t)$ along the ODE-induced trajectory. Colors show log-frequency over integration steps aggregated across runs.

**Per-task Accuracy under Heterogeneous Task Utility**. To further characterize behavior under our utility-weighted setting, we report the per-task accuracies on the full 20-task stream under heterogeneous task utility. The results are shown in Fig. 8.

**Path Property via Two-Task Trajectory Sweeps**. To visualize the controllability induced by our ODE formulation, we study a two-task setting and evaluate models obtained at different terminal times along the trajectory. Given endpoints $\theta_0$ (current) and $\theta_1$ (incoming), we integrate the rectified ODE and record checkpoints $\theta(t)$ at $t \in \{0, 0.1, \ldots, 1\}$. For each checkpoint, we report the per-task accuracy on the two corresponding test sets and their mean, as shown in Table 6.

**Statistics of the Rectification Coefficient** $\gamma$. To better understand how often the proposed dynamics actively suppress loss-increasing motion, we summarize the empirical behavior of the adaptive coefficient $\gamma(\theta, t)$ along the ODE trajectory. Fig. 5 reports a 2D histogram of $\gamma$ versus integration time $t$, aggregated over all Euler steps across runs, with color indicating log-frequency. The mass concentrates near $\gamma \approx 1$ over a large portion of the trajectory, suggesting that the rectification is typically non-invasive and the update can follow the geometric transport direction without attenuation. Smaller values of $\gamma$ appear in localized regions, indicating that damping is applied selectively when the gradient-aligned component would otherwise increase the loss, which is consistent with our design goal of regulating barrier-forming motion while preserving overall progress toward the target model.

**Alignment Ratio Between** $u_t$ **and the Loss Gradient**. We further examine the geometric condition used in our convergence analysis by measuring the relative magnitude of the gradient-aligned component in the base field. Fig. 6 visualizes the empirical distribution of the ratio $\|\mathcal{P}_{g_t}(u_t)\|/\|u_t\|$ as a function of integration time $t$, aggregated over all Euler steps across runs, with color indicating log-frequency. The distribution remains largely concentrated below moderate values throughout the trajectory, suggesting that $u_t$ is typically not dominated by its projection onto the loss gradient. This observation is consistent with our assumption that the motion toward the target generally has a substantial tangent component, and it helps explain why the rectified dynamics can both regulate loss-increasing directions and still maintain stable progression along the trajectory.

**Empirical Validation of Time Scheduling.** We empirically investigate whether the best operating point along the ODE-induced trajectory reflects the expected accumulation of historical knowledge in continual merging. Intuitively, as more task models have been merged, the current model $\Psi_{k-1}$ contains an increasingly larger amount of accumulated knowledge, and the next update should therefore retain a larger fraction of $\Psi_{k-1}$ while assigning a smaller relative contribution to the incoming model $\psi_k$. To examine this behavior, at each merge step we construct the trajectory between $\Psi_{k-1}$ and $\psi_k$, evaluate candidate merged models at uniformly sampled operating times $t \in [0, 1]$ with interval $0.05$, and select the

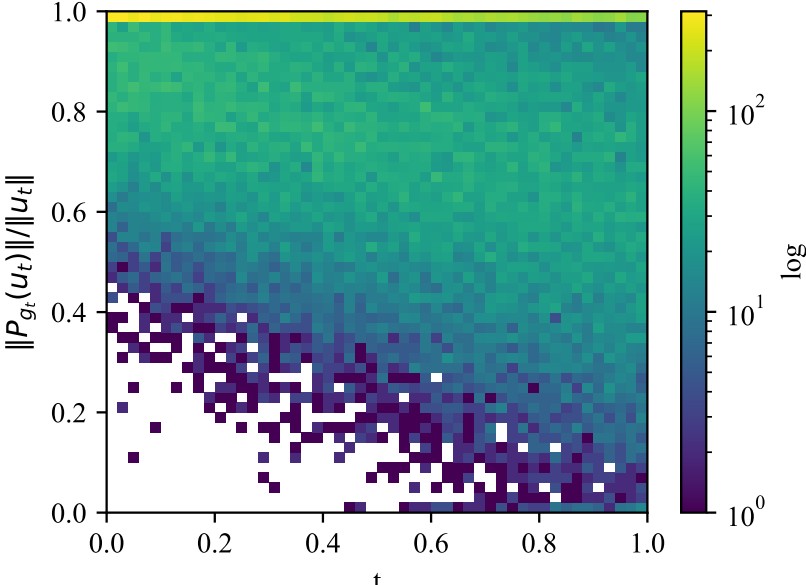

*Figure 6.* Empirical statistics of the alignment ratio $\|\mathcal{P}_{g_t}(u_t)\|/\|u_t\|$ along the ODE trajectory. Colors show log-frequency over integration steps aggregated across runs.

empirically best operating time $t_k^\star$ according to validation performance. Since $t_k^\star$ controls the degree of movement toward the incoming model, $1 - t_k^\star$ can be interpreted as a trajectory-level proxy for the retained contribution of the previous merged model. We then compute the Spearman rank correlation between $\{1 - t_k^\star\}$ and the equal-utility historical retention ratio $\{(k-1)/k\}$. As shown in Table 7, the retained historical contribution implied by the empirically selected operating times is strongly correlated with the historical retention ratio across architectures and stream lengths. This indicates that the best operating points naturally preserve a larger fraction of the accumulated model as the stream grows, which supports the diminishing incoming-task schedule $t_k = 1/k$ in the equal-utility setting and its utility-aware extension $t_k = w_k / \sum_{i=1}^{k} w_i$.

## C. ODE-based Merging Procedure

Algorithm 1 summarizes the complete procedure of ODE-M for continual model merging. At merge step $k$, we take the currently deployed model $\Psi_{k-1}$ as the start point $\theta_0$ and the incoming task-adapted model $\psi_k$ as the target $\theta_1$. We then construct an update trajectory by integrating an ODE induced by the rectified velocity field in Section 3.2. Specifically, at each integration time $t$, we form the base transport field $u_t(\theta) = \alpha(t)(\theta_1 - \theta)$ and compute the calibration gradient $g_t = \nabla_\theta \mathcal{L}(\theta; \mathcal{D}_{\text{cal}})$ on a small calibration set $\mathcal{D}_{\text{cal}}$. We decompose $u_t$ into a gradient-aligned component $u_\parallel$ and an orthogonal component $u_\perp$, and modulate the loss-increasing motion by an adaptive factor $\gamma(\theta, t) \in [0, 1]$. This yields the rectified field $v_t(\theta) = u_\perp + \gamma(\theta, t) u_\parallel$, whose induced trajectory explicitly regulates loss-increasing updates while preserving geometric progress toward $\theta_1$.

To obtain the next merged model, we do not necessarily integrate to $t = 1$. Instead, we select an operating time $t_k$ along the trajectory and set $\Psi_k = \theta(t_k)$, where $t_k$ follows the diminishing schedule in Section 3.4 (e.g., $t_k = 1/k$). In our implementation, we use Euler integration with step size $h$ and terminate once $t$ reaches $t_k$. The computational cost of each merge is thus dominated by $O(\lceil t_k/h \rceil)$ first-order gradient evaluations on $\mathcal{D}_{\text{cal}}$, which keeps merging lightweight compared to full retraining. Unless otherwise stated, we set $|\mathcal{D}_{\text{cal}}| = 1024$ and $h = 0.05$.

*Table 6.* Per-task accuracy along the ODE-induced trajectory $\theta(t)$ connecting two task-adapted models. $t = 0$ corresponds to the start model $\theta_0$ and $t = 1$ to the target model $\theta_1$. **Avg** denotes the mean accuracy of the two tasks. For each task pair, the best **Avg** along the trajectory is highlighted in bold.

| Pair | Metric | 0.0 | 0.1 | 0.2 | 0.3 | 0.4 | 0.5 | 0.6 | 0.7 | 0.8 | 0.9 | 1.0 |
|---|---|---|---|---|---|---|---|---|---|---|---|---|
| | cars | 0.785 | 0.788 | 0.782 | 0.772 | 0.787 | 0.763 | 0.698 | 0.658 | 0.605 | 0.541 | 0.472 |
| cars → resisc45 | resisc45 | 0.513 | 0.691 | 0.756 | 0.833 | 0.936 | 0.939 | 0.938 | 0.946 | 0.950 | 0.951 | 0.951 |
| | **Avg** | 0.649 | 0.739 | 0.769 | 0.803 | 0.821 | **0.851** | 0.818 | 0.802 | 0.777 | 0.746 | 0.712 |
| | sun397 | 0.749 | 0.752 | 0.750 | 0.743 | 0.749 | 0.731 | 0.683 | 0.653 | 0.609 | 0.554 | 0.490 |
| sun397 → eurosat | eurosat | 0.467 | 0.704 | 0.865 | 0.950 | 0.971 | 0.979 | 0.983 | 0.987 | 0.988 | 0.990 | 0.991 |
| | **Avg** | 0.608 | 0.728 | 0.808 | 0.847 | 0.850 | **0.855** | 0.833 | 0.820 | 0.798 | 0.772 | 0.740 |
| | dtd | 0.797 | 0.798 | 0.785 | 0.784 | 0.774 | 0.751 | 0.626 | 0.557 | 0.487 | 0.418 | 0.359 |
| dtd → resisc45 | resisc45 | 0.377 | 0.544 | 0.692 | 0.802 | 0.874 | 0.935 | 0.935 | 0.945 | 0.951 | 0.953 | 0.951 |
| | **Avg** | 0.587 | 0.671 | 0.739 | 0.793 | 0.824 | **0.843** | 0.781 | 0.751 | 0.719 | 0.685 | 0.655 |
| | gtsrb | 0.989 | 0.988 | 0.986 | 0.983 | 0.976 | 0.962 | 0.878 | 0.728 | 0.541 | 0.357 | 0.243 |
| gtsrb → resisc45 | resisc45 | 0.205 | 0.346 | 0.522 | 0.695 | 0.811 | 0.930 | 0.931 | 0.939 | 0.946 | 0.950 | 0.951 |
| | **Avg** | 0.597 | 0.667 | 0.754 | 0.839 | 0.893 | **0.946** | 0.904 | 0.834 | 0.743 | 0.653 | 0.597 |
| | gtsrb | 0.989 | 0.988 | 0.986 | 0.982 | 0.971 | 0.962 | 0.868 | 0.745 | 0.571 | 0.404 | 0.273 |
| gtsrb → svn | svn | 0.411 | 0.564 | 0.713 | 0.832 | 0.905 | 0.950 | 0.960 | 0.969 | 0.972 | 0.973 | 0.973 |
| | **Avg** | 0.700 | 0.776 | 0.850 | 0.907 | 0.938 | **0.956** | 0.914 | 0.857 | 0.771 | 0.689 | 0.623 |

*Table 7.* Empirical validation of the time-scheduling behavior. We report the Spearman rank correlation between the retained historical contribution $\{1 - t_k^\star\}$ implied by the empirically selected operating times and the equal-utility historical retention ratio $\{(k-1)/k\}$.

| Architecture | 8 tasks | 14 tasks | 20 tasks |
|---|---|---|---|
| ViT-B/32 | 0.941 | 0.907 | 0.884 |
| ViT-B/16 | 0.953 | 0.921 | 0.896 |
| ViT-L/14 | 0.936 | 0.912 | 0.889 |

# D. Baseline Details under Heterogeneous Task Utility

We adapt the baseline methods to the heterogeneous task utility setting in Section 4.3. At the $k$-th merge step, the system has observed task-adapted models $\{\psi_i\}_{i=1}^k$ with non-negative task utilities $\{w_i\}_{i=1}^k$. We denote the accumulated utility by

$$W_k \triangleq \sum_{i=1}^{k} w_i, \tag{43}$$

and use the incoming-task utility ratio

$$\eta_k \triangleq \frac{w_k}{W_k}. \tag{44}$$

When all tasks have equal utilities, this reduces to the standard uniform continual schedule $\eta_k = 1/k$.

For a fair comparison, all utility-aware baselines are adapted under the same online CMM protocol: at step $k$, the method updates the current merged model $\Psi_{k-1}$ using only the incoming task model $\psi_k$, the shared pre-trained model $\psi_0$, and the utility ratio $\eta_k$. Whenever a closed-form prefix expression is shown, it is used only to clarify the induced task contribution; the implementation follows the corresponding recursive online update.

**Continual Averaging.** For the averaging baseline, the utility-aware update is the natural recursive weighted average:

$$\Psi_k^{\text{C.SWA}} = (1 - \eta_k)\Psi_{k-1}^{\text{C.SWA}} + \eta_k \psi_k. \tag{45}$$

Equivalently, this recursion yields

$$\Psi_k^{\text{C.SWA}} = \sum_{i=1}^{k} \frac{w_i}{W_k} \psi_i, \tag{46}$$

which recovers the usual uniform average when $w_1 = \cdots = w_k$.

---

**Algorithm 1** Barrier-Aware ODE for Continual Model Merging (ODE-M)

---

**Require:** Stream of task-adapted models $\{\psi_k\}_{k=1}^K$, pretrained weights $\psi_0$; calibration set $\mathcal{D}_{\mathrm{cal}}$; loss $\mathcal{L}(\theta; \mathcal{D}_{\mathrm{cal}})$; step size $h$; scheduler $\alpha(t)$; time rule $t_k$ (e.g., $t_k = 1/k$).

**Ensure:** Merged model $\Psi_K$.

1:   $\Psi_1 \leftarrow \psi_1$
2: **for** $k = 2 \to K$ **do**
3:     $\theta_0 \leftarrow \Psi_{k-1}, \ \theta_1 \leftarrow \psi_k$
4:     $\theta \leftarrow \theta_0, \ t \leftarrow 0$
5:     **while** $t < t_k$ **do**
6:       $g \leftarrow \nabla_\theta \mathcal{L}(\theta; \mathcal{D}_{\mathrm{cal}})$
7:       $u \leftarrow \alpha(t) \, (\theta_1 - \theta)$
8:       $u_\| \leftarrow \frac{\langle u, g \rangle}{\|g\|^2} \, g, \quad u_\perp \leftarrow u - u_\|$
9:       $\Delta\mathcal{L} \leftarrow \mathcal{L}(\theta_1; \mathcal{D}_{\mathrm{cal}}) - \mathcal{L}(\theta_0; \mathcal{D}_{\mathrm{cal}})$
10:      **if** $\langle g, u \rangle \le 0$ **then**
11:        $\gamma \leftarrow 1$
12:      **else**
13:        $\gamma \leftarrow \mathrm{clip}\left( \frac{\Delta\mathcal{L}}{\langle g, u \rangle}, 0, 1 \right)$
14:      **end if**
15:      $v \leftarrow u_\perp + \gamma \, u_\|$
16:      $\theta \leftarrow \theta + h \, v$ {Euler update}
17:      $t \leftarrow t + h$
18:     **end while**
19:     $\Psi_k \leftarrow \theta$
20: **end for**
21: **return** $\Psi_K$

---

**Continual Task Arithmetic.** Continual Task Arithmetic operates on task vectors relative to the shared pre-trained model. Let

$$\delta_k \triangleq \psi_k - \psi_0 \tag{47}$$

be the incoming task vector, and let $\bar{\delta}_{k-1}^{\mathrm{TA}}$ denote the accumulated task vector from the previous step. We update the accumulated vector using the same utility ratio:

$$\bar{\delta}_k^{\mathrm{TA}} = (1 - \eta_k)\bar{\delta}_{k-1}^{\mathrm{TA}} + \eta_k \delta_k. \tag{48}$$

The merged model is then obtained by applying the original task-arithmetic scaling coefficient:

$$\Psi_k^{\mathrm{C.TA}} = \psi_0 + \lambda_{\mathrm{TA}} \bar{\delta}_k^{\mathrm{TA}}. \tag{49}$$

Here $\lambda_{\mathrm{TA}}$ is kept as the method-specific scaling coefficient used by Task Arithmetic. This distinction is important: when $\lambda_{\mathrm{TA}} = 1$, the update reduces to weighted averaging, but in general Continual Task Arithmetic remains a task-vector method rather than a checkpoint averaging method.

**Continual Ties-Merging.** Ties-Merging applies sparsification and sign-consistent merging to task vectors. To preserve this mechanism in the online setting, we adapt Ties-Merging as a sequential pairwise merge between the accumulated update and the incoming task vector. Let

$$\Delta_{k-1}^{\mathrm{TIES}} \triangleq \Psi_{k-1}^{\mathrm{C.TIES}} - \psi_0 \tag{50}$$

be the current accumulated update. At step $k$, we form two utility-weighted candidate vectors,

$$(1 - \eta_k)\Delta_{k-1}^{\mathrm{TIES}} \quad \text{and} \quad \eta_k \delta_k, \tag{51}$$

and feed them into the standard Ties-Merging operator:

$$\bar{\delta}_k^{\mathrm{TIES}} = \mathrm{TIES}\left( \left\{ (1 - \eta_k)\Delta_{k-1}^{\mathrm{TIES}}, \eta_k \delta_k \right\} \right). \tag{52}$$

The final merged model is

$$\Psi_k^{\text{C.TIES}} = \psi_0 + \lambda_{\text{TIES}} \bar{\delta}_k^{\text{TIES}}, \tag{53}$$

where the trimming ratio, sign-election rule, and scaling coefficient $\lambda_{\text{TIES}}$ follow the OPCM (Tang et al., 2025b) configuration. Thus, only the relative contribution of the previous and incoming updates is changed to match the heterogeneous utility schedule, while the core Ties-Merging procedure remains unchanged.

**OPCM.** OPCM is a projection-based continual merging method that sequentially incorporates each incoming task model by projecting its task vector against the current accumulated update. Let

$$\Delta_{k-1}^{\text{OPCM}} \triangleq \Psi_{k-1}^{\text{OPCM}} - \psi_0, \qquad \delta_k \triangleq \psi_k - \psi_0. \tag{54}$$

Following the original OPCM procedure, the incoming task vector is first projected with respect to the current accumulated update:

$$\hat{\delta}_k = \mathcal{P}_\alpha^{(k-1)} \left( \delta_k; \Delta_{k-1}^{\text{OPCM}} \right), \tag{55}$$

where $\mathcal{P}_\alpha^{(k-1)}(\cdot)$ denotes OPCM's orthogonal projection operator. In practice, this projection is applied to weight matrices in linear layers, while the remaining parameters follow the original OPCM treatment.

To adapt OPCM to heterogeneous task utility, we keep its projection and adaptive scaling mechanisms unchanged, and only replace the default uniform contribution of the incoming projected update with the utility ratio

$$\eta_k = \frac{w_k}{W_k}. \tag{56}$$

Specifically, we form the utility-aware accumulated update as

$$A_k^{\text{OPCM}} = (1 - \eta_k) A_{k-1}^{\text{OPCM}} + \eta_k \hat{\delta}_k, \tag{57}$$

where $A_{k-1}^{\text{OPCM}}$ denotes the accumulated projected update maintained by OPCM. The merged model is then obtained with OPCM's original adaptive scaling rule:

$$\Psi_k^{\text{OPCM}} = \psi_0 + \frac{A_k^{\text{OPCM}}}{\lambda_k^{\text{OPCM}}}. \tag{58}$$

Here $\lambda_k^{\text{OPCM}}$ is computed following the original OPCM scaling strategy, which controls the magnitude of the merged update relative to the pre-trained model. Thus, the utility-aware adaptation does not change OPCM's core orthogonal projection mechanism; it only adjusts the relative contribution of the incoming projected update so that the sequential update is aligned with the heterogeneous utility schedule.

Overall, these adaptations ensure that all baselines are evaluated under the same heterogeneous utility specification. Without this alignment, a method designed for a uniform macro-average objective may be artificially advantaged or penalized when the evaluation objective emphasizes a non-uniform task mixture.

# E. Discussions and Limitations

**Discussions**. Our method reframes a continual merging update as selecting an operating point along an ODE-induced connecting trajectory between the current deployed model and the incoming task-adapted model. This perspective makes the update *path-dependent* rather than *endpoint-only*, exposing an explicit control handle through time scheduling along the trajectory. Empirically, the resulting behavior is consistent with the intuition suggested by mode connectivity, namely that well-chosen continuous transitions can avoid sharp loss spikes even when direct interpolation is unfavorable. The rectified velocity field further provides a lightweight mechanism to regulate loss-increasing motion using only first-order feedback on a small calibration set, which keeps the procedure compatible with training-free continual merging pipelines.

**Limitations**. Our approach requires access to a loss function and first-order gradients on a calibration set during merging. While the calibration budget is small in our implementation, this assumption may not hold in fully black-box deployment scenarios, and the effectiveness can depend on how representative the calibration data is of previously merged tasks. In addition, the method introduces several numerical choices, such as the integration step size and the time schedule along the trajectory. We find the method robust across a wide range of these settings, but extremely coarse discretization or poorly chosen schedules can still degrade performance. Finally, although our experiments cover multiple CLIP ViT backbones and stream lengths, the current evaluation focuses on vision classification-style task streams; validating the same trajectory-based control principles in other modalities and larger-scale foundation models remains an important direction for future work.

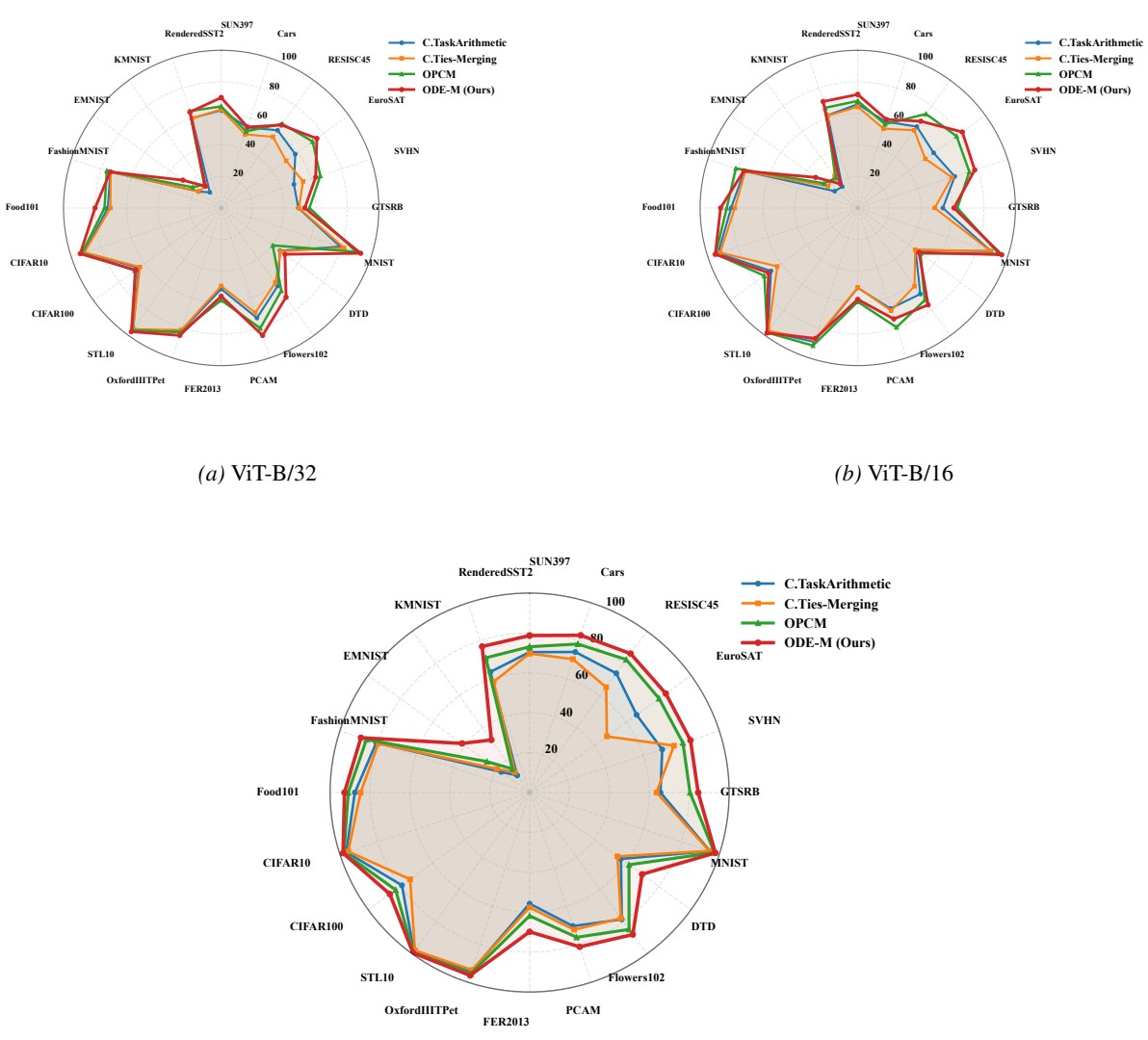

*(a)* ViT-B/32

*(b)* ViT-B/16

*(c)* ViT-L/14

*Figure 7.* **Per-task performance on 20-task continual merging.** Radar plots show the per-task accuracy of different continual merging methods on a 20-task stream for three CLIP ViT backbones: (a) ViT-B/32, (b) ViT-B/16, and (c) ViT-L/14.

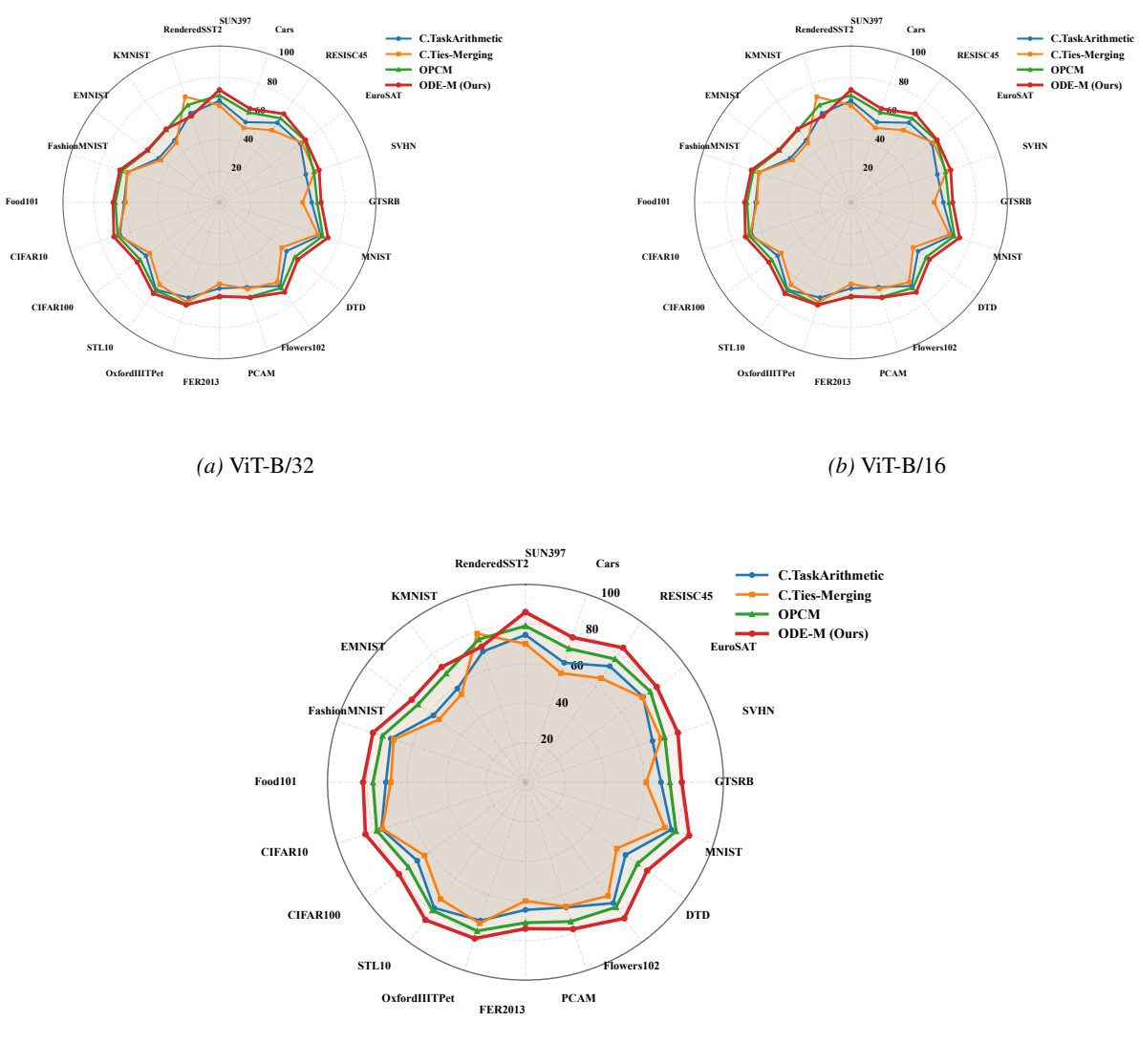

*(a)* ViT-B/32

*(b)* ViT-B/16

*(c)* ViT-L/14

*Figure 8.* **Per-task performance on 20-task continual merging (heterogeneous utility).** Radar plots show the per-task accuracy of different continual merging methods on a 20-task stream for three CLIP ViT backbones: (a) ViT-B/32, (b) ViT-B/16, and (c) ViT-L/14.

