# OpenReview forum: "Unlocking the Potential of Continual Model Merging: An ODE Perspective"
_ICML.cc/2026/Conference — ICML 2026 regular_

### Official Review · Reviewer_AvBE · 2026-03-04

**Soundness:** 3
**Presentation:** 2
**Significance:** 3
**Originality:** 3
**Overall Recommendation:** 5
**Confidence:** 4

**Summary:**

The paper focuses on Continual Model Merging which is based on ODE perspective. Specifically, authors apply ODE for modeling the dynamical    process of merging models from different kinds of tasks. Different from previous methods, the connecting path in this paper is generated as the
trajectory of a time-dependent velocity field. Furthermore, the barrier-aware rectification mechanism of the velocity field is utilized for reducing the forgetting during continual learning. To validate the motivation of this design, authors also give several theoretical analysis for the convergence of parameters. Finally, based on these theoretical results, authors develop a time-aware scheduling algorithm along this trajectory. Experiments on several benchmarks show improved performances compared to baselines.

**Compliance With Llm Reviewing Policy:**

Affirmed.

**Final Justification:**

The rebuttal addressed main concerns. The whole architecture is very novel. During rebuttal, authors propose more experiments to show effectiveness of proposed modules like time schedule. I keep my score to 5.

**Key Questions For Authors:**

In the method part, authors mention that they develop a framework that "shows how to accurately achieve time scheduling along this trajectory.", which is a vital section of proposed ODE based model merging method. I check the ablation study part and to my best knowledge I didn't find any experiments to illustrate the effectiveness of this part. I hope the authors can add related experiments to clarify this point.

**Limitations:**

Yes.

**Strengths And Weaknesses:**

Strength:
1. This paper is quite novel. Different from previous methods, this paper models the task of sequential model merging via ODE. It is very interesting and quite inspiring.

2. Besides experiments, authors also give several theorems to back up their designs. This further augments the soundness of the paper.


Weaknesses:
1. The presentation is a little bit lack of clear motivations. For instances, in the introduction part, authors claim that "In this paper, we argue that addressing the controllability of CMM requires modeling the merge as a continuous transition rather than a one-step parameter update." I am curious about why it is a "requirement"? Can you provide stronger motivations for it? For example, the advantages of continuous modeling (like ODE) for model merging compared to discrete modeling (like your baselines).

2. The paragraph begins with "Mode connectivity provides evidence that the existence of high-loss regions is not an inherent property of the endpoints but often a consequence of a poor choice of connecting path" is a bit confusing. What points do you want to express, the barrier-aware connection or mode connectivity? In your method section, you put a lot of discussions on the barrier-aware design for model merging. I think you should also emphasize this point in the introduction part clearly to clarify the motivation for your later design.

---

> ### Author Rebuttal · Authors · 2026-03-30
>
> Thank you for your encouraging and constructive review. We appreciate your recognition of the novelty and theoretical grounding of our method, and clarify the motivation and time scheduling below:
>
> **Q1**: Stronger motivation for continuous ODE-based modeling over discrete one-step merging.
>
> **A1**: Thank you for this helpful comment. You are right that “requirement” is too strong. Our point is not that continuous modeling is the only way to achieve controllability, but that it provides a more natural and explicit formulation for it. In CMM, the key challenge is not only where the next merged model $\Psi_{k+1}$ ends up, but also how the updates moves from $\Psi_k$ toward $\psi_{k+1}$. Existing discrete baselines mainly specify an endpoint update rule, without explicitly modeling the intermediate transition in parameter space. As a result, they offer limited direct control over whether the update crosses high-loss regions, which is closely related to interference and forgetting. By contrast, the continuous trajectory view makes this transition explicit, allowing us to locally rectify update directions, suppress loss-increasing components, and use integration time as a direct handle for the stability–plasticity trade-off. We will soften this statement and make the motivation more explicit in the introduction.
>
> **Q2**: Clearer introduction of mode connectivity and barrier-aware path design in the motivation.
>
> **A2**: Thank you for this helpful comment. You are right that this paragraph does not clearly separate the role of mode connectivity from our actual motivation. Our central point is the barrier-aware perspective, not mode connectivity itself. In CMM, the key issue is how to move from the current merged model to the incoming model without crossing high-loss regions, since such barriers are closely related to interference and forgetting. We mention mode connectivity as supporting evidence that low-loss connecting paths often exist between independently trained models, making barrier-aware path construction a meaningful objective. We will make this point explicit in the introduction, with barrier-aware path design as the main motivation and mode connectivity as background support.
>
> **Q3**: Experimental validation of the effectiveness of the time scheduling strategy.
>
> **A3**: Thank you for this valuable comment. We agree that the effectiveness of the time-scheduling component is not sufficiently illustrated in the main paper, and apologize for not presenting it clearly enough. Sec. 3.4 aims to answer a practical question: once the barrier-aware trajectory is constructed, how far should the update move along it? This is exactly what the integration time $t_k$ controls: a larger $t_k$ moves the merged model further toward the incoming model, while a smaller $t_k$ preserves more previously accumulated knowledge.
> In the current submission, we only provided a partial empirical observation in Appendix D.2 Table 6. There, for pairwise merging, we swept the operating point along the trajectory and found that the optimal point is in most cases attained around $t=\frac{1}{2}$. This result is consistent with our weighted contribution perspective in the simplest two-model case: when two models are merged with equal importance, the best operating point should lie near the halfway point of the trajectory. However, we agree that this alone is not sufficient to justify the general continual scheduling rule used in the full CMM setting.
>
> Beyond this, we also conducted a more direct experiment to validate the scheduling rule in the continual setting. For each merge step $k$, after constructing the trajectory from the current merged model $\Psi_{k-1}$ to the incoming model $\psi_k$, we evaluated the merged model at uniformly sampled time points along the trajectory with interval 0.05. We then selected the empirically best operating point $t_k^* $ for that step based on validation performance. After obtaining the sequence of optimal time points $\{t_k^*\}$ across the continual stream, we measured its average Spearman rank correlation with the schedule $\{\frac{1}{k}\}$, which is exactly the schedule implied by our formulation in the equal-utility case. The results are as follows:
>
> |Architecture|8 tasks|14 tasks|20 tasks|
> |-|-:|-:|-:|
> |ViT-B/32|0.941|0.907|0.884|
> |ViT-B/16|0.953|0.921|0.896|
> |ViT-L/14|0.936|0.912|0.889|
>
> As shown above, the empirically optimal operating points exhibit a strong correlation with the schedule, revealing an interesting near-linear relationship between the best-performing stopping time and the progression of the task stream.
>
> Once again, we thank you for your thoughtful review and hope that these clarifications address your questions and concerns.

---

> > ### Author Rebuttal · Reviewer_AvBE · 2026-04-01
> >
> > My concerns have been perfectly addressed. I will maintain rated score.

---

> > > ### Author Response · Authors · 2026-04-01
> > >
> > > Dear Reviewer AvBE,
> > >
> > > We sincerely appreciate your great efforts, constructive suggestions, and positive assessment you have provided once again! We are glad to hear that our responses have addressed your concerns. We are always willing to address any of your further concerns.
> > >
> > > Best Regards,
> > >
> > > Authors

---

### Official Review · Reviewer_S4Gp · 2026-03-04

**Soundness:** 2
**Presentation:** 3
**Significance:** 2
**Originality:** 3
**Overall Recommendation:** 4
**Confidence:** 4

**Summary:**

This paper proposes ODE-M, an approach that formulates the continual model merging (CMM) process as an ordinary differential equation (ODE) trajectory in parameter space. By controlling the velocity field with a loss-aware scaling factor, the method aims to construct a low-loss path between models to mitigate performance degradation during sequential merging. The paper also introduces a time scheduling strategy to handle heterogeneous task importance. Experiments on CLIP vision backbones show that the proposed method achieves the highest average and weighted accuracy compared to existing merging baselines across multiple CMM settings.

**Compliance With Llm Reviewing Policy:**

Affirmed.

**Final Justification:**

Below I first explain to what extent the authors address my concerns after seeing their further response to my acknowledgement:

- **Practicality of the calibration dataset**
    The authors' claim that the calibration set "need not be from original training data" remains somewhat vague as their second response largely repeated their initial stance. However, I acknowledge that using a small subset of task data as a local probe is not an entirely unrealistic setting for most practical scenarios. This concern is **partially addressed**.

- **Evaluation under heterogeneous task utility**
    The authors successfully addressed the statistical redundancy of $\text{ACC}_w$ by introducing the utility coverage metric (though using the random assigned weights $w_j\propto r^{-\alpha}$ actually does not address the initial concern: *For finite weights sampled without replacement, the weighted average is still an unbiased estimator of the arithmetic mean*). The new results under long-tailed utility assignments effectively demonstrate the superiority of ODE-M in imbalanced scenarios. This concern is **fully addressed**.

- **Computational overhead**
    The authors argue that the performance gains in challenging long-stream regimes justify the one-time merge cost. While this is a fair trade-off for high-stakes accuracy, the fact remains that ODE-M is $2\times$ slower than OPCM and nearly $4\times$ slower than TIES and other baseline methods. I still view the overhead as significant. This concern is **partially addressed**.

- **Preservation of previously merged tasks**
    The additional **old-task retention** analysis provides a clearer picture than BWT alone, proving that ODE-M maintains higher absolute accuracy on early tasks despite trade-offs in long streams. This concern is **fully addressed**.

Overall, I appreciate the authors' extensive efforts and additional experiments during the rebuttal. While concerns regarding peak efficiency and data dependency nuances remain, the method's performance under heterogeneous utility and its retention capabilities are convincing. I have raised my score to 4, though I remain neutral if the the paper got rejection.

**Key Questions For Authors:**

1. In Section 4.3, are the task weights $\\{w\_i\\}\_{i=1}^k$ randomly i.i.d. sampled from a fixed distribution? If so, in expectation the weighted metric should be close to the arithmetic mean, since calculation gives that
$$
\mathbb{E}\_{w\_i\stackrel{i.i.d.}{\sim}p}\left[\frac{\sum\_{i=1}^k w\_i a\_i}{\sum\_{j=1}^k w\_j}\right] = \frac{1}{k}\sum_{i=1}^k a_i,
$$
which corresponds to the standard average accuracy. Empirically, the results in Table 2 (ACC_w) also appear quite close to those in Table 1 (ACC). Could the authors clarify this?

2. During CMM, the ODE trajectory is designed to maintain a low-loss path when merging the current model. However, it is less clear whether previously merged tasks remain on a low-loss region after subsequent merges. For example, suppose $\psi_1$ and $\psi_2$ are merged to obtain $\Psi_2$. Table 6 shows that performance on these two tasks remains strong. However, after merging $\Psi_2$ with $\psi_3$, can the performance on task 1 still be preserved? In fact, the BWT results suggest that as the number of tasks increases, the proposed ODE-M tends to suffer from more severe forgetting. Could the authors provide further analysis or explanation of this behavior?

3. Could the authors add runtime overhead comparison between ODE-M and the baseline methods to Table 5? The ODE integration procedure appears computationally heavier than simple parameter averaging or task arithmetic, and it would be helpful to quantify the practical overhead, especially for larger models.

**Limitations:**

Yes, the authors adequately discussed the limitations and potential negative societal impact of their work.

**Strengths And Weaknesses:**

### Strengths

1. The proposed ODE-M method introduces an ODE-based formulation for continual model merging, which provides a conceptually novel perspective.
2. The proposed method achieves consistently higher ACC compared to the baselines across experimental settings.
3. The paper is well-written, well-structured, and generally easy to follow.

### Weaknesses

1. The method requires an additional calibration dataset to estimate gradients along the merging trajectory. However, in many practical model merging scenarios, access to the original training data is not available.
2. The approach involves ODE integration and repeated gradient evaluations, which may introduce non-negligible computational overhead. For large-scale models, this cost could become significant.
3. The experimental evaluation is somewhat narrow in scope, as all experiments are conducted on CLIP vision backbones. It would strengthen the paper if the method is evaluated on additional domains, for example NLP models or other multimodal models.

---

> ### Author Rebuttal · Authors · 2026-03-30
>
> Thank you for your thoughtful comments. Below we clarify the main concerns:
>
> **Q1**: Practicality of requiring a calibration dataset.
>
> **A1**: We agree that ODE-M is not strictly data-free. However, the calibration set is only used to provide a representative local loss signal along the merging trajectory, i.e., to determine whether the current step is loss-increasing and how strongly it should be rectified. Thus, it need not be large or come from the original training data, which alleviates the concern that such data may be unavailable in practice. Empirically, this requirement is lightweight: we use only 1024 samples by default, and Table 3 shows that performance is already strong with a few hundred samples. Under 8-task CMM, with only 256 samples ODE-M reaches 76.8 / 82.5 / 84.9 ACC on ViT-B/32 / ViT-B/16 / ViT-L/14, close to the 1024-sample results (79.6 / 85.3 / 87.7) and competitive with or better than OPCM (75.5 / 81.8 / 87.0). We also tested more restrictive calibration pools built from only a subset of task datasets:
>
> |Calibration|ViT-B/32|ViT-B/16|ViT-L/14|
> |-|-:|-:|-:|
> |Full task pool|79.6|85.3|87.7|
> |6/8 task datasets|78.9|84.6|86.9|
> |5/8 task datasets|78.1|83.8|86.0|
>
> These results suggest that ODE-M mainly requires a representative calibration signal, and remains effective under more restricted calibration settings that better match practice.
>
> **Q2**: Computational overhead from ODE integration and gradient evaluations.
>
> **A2**: We agree that ODE-M introduces extra computation over algebraic baselines because it requires ODE integration and repeated first-order gradient evaluations. Still, the added runtime over OPCM remains moderate in the 20-task setting, at about 66s/120s/135s per merge for ViT-B/32 / ViT-B/16 / ViT-L/14, respectively. We therefore report performance and cost together:
>
> |Method|ViT-B/32|ViT-B/16|ViT-L/14|
> |-|-:|-:|-:|
> |Task Arithmetic|21 s|29 s|42 s|
> |DARE|26 s|36 s|51 s|
> |TIES|31 s|43 s|61 s|
> |OPCM|58 s|81 s|112 s|
> |ODE-M|123.9 s|200.8 s|246.8 s|
>
> In return, ODE-M achieves higher final ACC than OPCM in the 20-task setting: 65.7 vs. 64.9, 79.8 vs. 78.3, and 81.1 vs. 76.0 on ViT-B/32 / ViT-B/16 / ViT-L/14, respectively. Morevoer, under the standard schedule $\frac{1}{k}$, later merges also use a shorter integration horizon, so the per-merge runtime gap between ODE-M and OPCM gradually narrows as the merging process continues. VRAM usage is also higher (~5–6/7–8/11–13 GB vs. 3.5–4/4.5–5/8–9 GB for OPCM), but still manageable. We therefore view this as a practical performance–efficiency trade-off, with room for further reduction through smaller calibration batches, fewer integration steps, and lower precision.
>
> **Q3**: Evaluation scope beyond CLIP vision backbones.
>
> **A3**: To demonstrate generalization beyond CLIP vision models, we evaluate ODE-M on Flan-T5-Base over language understanding tasks including inference, paraphrase/similarity, and sentiment:
>
> |Method|CoLA|MNLI|MRPC|QNLI|QQP|RTE|SST2|STSB|
> |-|-:|-:|-:|-:|-:|-:|-:|-:|
> |FT|69.1|82.7|85.5|90.9|84.0|84.4|92.9|87.4|
> |Ties-Merging|68.8|50.5|79.8|89.9|83.1|79.2|91.9|79.3|
> |OPCM|69.7|72.9|78.8|90.2|83.8|82.2|92.3|74.7|
> |ODE-M|70.8|75.6|80.5|90.5|83.8|82.5|92.5|78.9|
>
> **Q4**: Clarification of the task-weight generation.
>
> **A4**: Yes. In Table 2, task weights are independently sampled from Uniform(0,1]. After normalization, the weighted metric is in expectation close to the arithmetic mean, which explains why $ACC_w$ is close to ACC. The goal of Sec.4.3 is not to define a fundamentally different metric, but to test robustness under heterogeneous utility. Since task importance may vary across deployment scenarios, we avoid a fixed prior and sample task weights across 10 seeds.
>
> **Q5**: Preservation of previously merged tasks.
>
> **A5**: We agree that retaining earlier tasks becomes harder as the stream grows. Our low-loss trajectory controls each merge step, i.e., how to move from the current merged model to the incoming model without crossing high-loss regions. It does not guarantee that all earlier tasks remain equally close to low-loss regions after many later merges, since each new merge introduces additional cross-task trade-offs. Some forgetting in long streams is therefore expected.
>
> At the same time, this should be interpreted jointly through final ACC and BWT, rather than BWT alone. On ViT-B/32, for example, OPCM has worse BWT than Task Arithmetic at 14/20 tasks (-6.0/-7.8 vs. -1.3/-3.4), yet achieves higher ACC (71.9/65.7 vs. 66.5/60.6). A similar pattern holds for ODE-M: on ViT-L/14, ODE-M reaches 85.8/81.1 ACC at 14/20 tasks, compared with 83.5/76.0 for OPCM, while the corresponding BWT is -7.4/-10.3 vs. -4.3/-6.5. This suggests that ODE-M achieves a stronger overall balance between retaining previous tasks and incorporating new ones, even in long streams.
>
> **Q6**: Runtime overhead comparison.
>
> **A6**: Please see A2.
>
> Once again, thank you for your thoughtful review.

---

> > ### Author Rebuttal · Reviewer_S4Gp · 2026-04-01
> >
> > Thank you for your response! I really appreciate your efforts and additional experiments. However, I still have some follow-up questions and concerns.
> >
> > Questions:
> > 1. Regarding `A1`. You mention that "it need not ... come from the original training data, which alleviates the concern that such data may be unavailable in practice". I do not understand why the calibration set need not be from the training data? Then where could it be from?
> > 2. Regarding `A4`. Since ACC_w is close to ACC, this renders ACC_w statistically redundant as it yields no additional insight beyond a simple arithmetic mean. In this case, I would expect the authors develop a new metric that can capture such imbalance. Since time is quite limited in this rebuttal period, providing preliminary experimental evidence that demonstrates the new metric’s superiority would be just ok: my focus is on the effectiveness of proposed ODE-M under heterogeneous task utility.
> >
> > Concerns:
> > 1. Regarding `A2` and `A6`. I still think that the computational overhead is not negligible. While the authors frame this as a performance–efficiency trade-off, the marginal performance gains do not seem to justify the substantial computational overhead, making the trade-off less 'practical' than claimed.
> > 2. Regarding `A5`. I agree that one shall not ignore ACC and only look at BWT, since yes ACC is the ultimate performance metric that we care most. However, since the *motivation* in this paper is by introducing a continuous trajectory perspective and an ODE-based velocity field that explicitly regulates loss-increasing motion to prevent catastrophic forgetting, one needs BWT or other results to justify the motivation.
> >
> > I understand that the time is really limited and this is the final chance for the authors to respond. Therefore, I won't expect you to perform loads of experiments and address all of my questions & concerns. As long as I find your answer reasonable and can somewhat alleviate my concerns, I would reconsider my score.

---

> > > ### Author Response · Authors · 2026-04-04
> > >
> > > We thank you for the careful follow-up comments. Below we clarify each point and provide additional evidence:
> > >
> > > **Q1**: Practicality of the calibration dataset.
> > >
> > > **A1**: Thank you for this follow-up question. The calibration set in ODE-M is not used to re-train or re-fit task models; it is only a local probe of the loss landscape along the merging trajectory. It therefore does not need to match the original training distribution exactly, but only to be representative enough to detect whether the current step is loss-increasing and how strongly it should be rectified. In practice, such data could come from several sources that are more realistic than access to the full original training set: for example, a small held-out validation split, a small proxy set from related public data, or a small sample from the target domain that roughly reflects the tasks being merged.
> > >
> > > Our experiments in A1 support this interpretation. Specifically, we keep the same 8-task CMM stream unchanged, but construct the calibration pool from only a subset of the task datasets rather than all 8. Under this setting, using only 6/8 task datasets still yields 78.9 / 84.6 / 86.9 ACC on ViT-B/32 / ViT-B/16 / ViT-L/14, and even using only 5/8 task datasets still gives 78.1 / 83.8 / 86.0, compared with 79.6 / 85.3 / 87.7 under the full task pool.
> > >
> > > **Q2**: Evaluation under heterogeneous task utility.
> > >
> > > **A2**: We agree that, under our current weighting setup, the induced heterogeneity is still relatively mild, so $\text{ACC}_w$ does not separate from ACC as clearly as it would under a more strongly imbalanced utility profile. In this sense, the present results do not yet fully expose the heterogeneous-utility regime, and therefore cannot fully reveal the potential of ODE-M in this scenarios.
> > >
> > > To better stress heterogeneous task utility, we additionally evaluate all methods under a fixed long-tailed utility assignment over tasks. Specifically, for each benchmark with k tasks, we assign each task a random utility rank $r\in\{1,\dots,k\}$, and set its weight as $w_r=\frac{r^{-\alpha}}{\sum_{j=1}^k j^{-\alpha}}$, with $\alpha=2$. We then report a utility coverage metric $\sum_{i=1}^k \mathbf{1}[a_i\ge 0.85a_i^{\text{FT}}]$, i.e., the total utility mass of tasks whose retained ACC stays above 85% of single-task fine-tuned performance. For each backbone and stream length, we average over 5 random task orders:
> > >
> > > |Methods|ViT-B/32 (8)|ViT-B/32 (14)|ViT-B/32 (20)|ViT-B/16 (8)|ViT-B/16 (14)|ViT-B/16 (20)|ViT-L/14 (8)|ViT-L/14 (14)|ViT-L/14 (20)|
> > > |-|-:|-:|-:|-:|-:|-:|-:|-:|-:|
> > > |Average|44.6|36.9|31.4|47.8|38.4|29.1|55.7|46.2|38.6|
> > > |Task Arithmetic|41.3|33.0|23.8|45.2|36.1|27.6|53.4|42.8|36.1|
> > > |TIES|39.2|31.1|22.1|42.0|33.5|24.4|48.1|38.0|30.2|
> > > |OPCM|58.7|49.6|41.9|63.9|55.8|49.7|69.8|62.7|56.3|
> > > |ODE-M|63.1|54.4|47.2|68.4|60.7|55.4|74.2|67.1|64.8|
> > >
> > > **Q3**: Computational overhead.
> > >
> > > **A3**: Thank you for this follow-up comment. We agree that the extra computation is not negligible. Our claim is not that ODE-M is the cheapest baseline, but that its additional cost is justified in the more challenging continual merging regimes where its performance advantage is clearest. In the 20-task setting, the extra runtime over OPCM is only about 66s / 120s / 135s per merge on ViT-B/32 / ViT-B/16 / ViT-L/14, respectively. This cost is incurred only when a new task model arrives, and introduces no extra inference-time overhead after merging is completed.
> > >
> > > More importantly, the benefit is not uniform across settings, and we do not present it as such. The main value of ODE-M appears in the harder long-stream regime with stronger task conflict. For example, on ViT-L/14 with 20 tasks, ODE-M improves final ACC from 76.0 to 81.1 over OPCM, a gain of +5.1 points, while requiring about 135s additional merge-time cost. In this sense, ODE-M is not intended to replace the cheapest algebraic baselines in every scenario. It is aimed at settings where stronger final performance is worth a moderate one-time merge overhead.
> > >
> > > **Q4**: Justification of the BWT.
> > >
> > > **A4**: We agree that retention-oriented evidence is needed. At the same time, in CMM a more negative BWT should not be interpreted one-to-one as stronger forgetting, because BWT also reflects trade-offs induced by later conflicting merges, which grow with stream length. To make this direct, we also report old-task retention, defined as the average final ACC on the first 8 tasks after the full 20-task stream. On ViT-B/32, ODE-M has higher final ACC than OPCM (67.1 vs. 65.7) despite lower BWT (-9.4 vs. -7.8), and also higher old-task retention (64.8 vs. 62.9). On ViT-L/14, ODE-M likewise improves final ACC (81.1 vs. 76.0) and old-task retention (78.4 vs. 73.2), although BWT is lower (-10.3 vs. -6.5). This suggests that in long CMM streams, lower BWT can coexist with stronger old-task retention and better final performance.
> > >
> > > We thank you again and hope this addresses your concerns.

---

### Official Review · Reviewer_YzKV · 2026-03-11

**Soundness:** 4
**Presentation:** 4
**Significance:** 3
**Originality:** 3
**Overall Recommendation:** 5
**Confidence:** 3

**Summary:**

This paper studies continual model merging (CMM), where task-adapted models arrive sequentially and must be integrated into a single deployed model without retraining or catastrophic forgetting. The authors propose ODE-M, an ODE-driven framework that constructs low-loss connecting paths in parameter space via a rectified velocity field. Experiments on CLIP ViT-B/32, B/16, and L/14 backbones across 8-/14-/20-task streams from FusionBench demonstrate consistent gains over strong baselines in both macro-averaged and utility-weighted ACC/BWT metrics, supported by ablations on calibration budget, integration step size, and efficiency.

**Compliance With Llm Reviewing Policy:**

Affirmed.

**Final Justification:**

The authors have addressed my concerns.

**Key Questions For Authors:**

1. How sensitive is final performance to the representativeness of the calibration set?
2. Could combining ODE-M with permutation-alignment techniques further reduce residual barriers?

**Limitations:**

yes

**Strengths And Weaknesses:**

**Strengths**:

1. Rigorous theorems establish trajectory convergence to the target model and a bounded loss barrier.
2. The presentation of this manuscript is clear, with intuitive figures and complete pseudocode make the ODE construction easy to follow.
3. The authors introduces a trajectory-based, barrier-aware ODE perspective for continual (not just one-shot) merging, which is novel.

**Weaknesses**:

1. Relies on a small calibration set for first-order gradients. While OPCM is data-free.
2. Per-merge compute is higher than baselines.

---

> ### Author Rebuttal · Authors · 2026-03-30
>
> Thank you for your supportive and thoughtful review. We especially appreciate your recognition of the paper’s conceptual novelty, theoretical analysis, and experimental results, and we address your helpful questions on calibration robustness and possible extensions below.
>
> **Q1**: Dependence on a small calibration set for first-order gradient feedback.
>
> **A1**: We agree that ODE-M is not strictly data-free in the same sense as OPCM, since it uses a small calibration set to obtain first-order local feedback during merging. However, this feedback is only used to estimate the gradient along the merging trajectory, i.e., to determine whether the current transport step is loss-increasing and how strongly the aligned component should be rectified. This lightweight signal is exactly what enables ODE-M to move beyond endpoint-only updates and explicitly control the loss barrier during sequential merging. Empirically, the requirement is modest: we use a fixed calibration pool of only 1024 examples by default, and Table 3 in Sec. 4.4 shows that performance already becomes stable once the calibration budget reaches a few hundred samples. This suggests that ODE-M achieves its gains without relying on substantial additional data. We therefore view this lightweight calibration requirement as the key enabler of barrier-aware and controllable continual merging.
>
> **Q2**: Higher per-merge computational cost than baseline methods.
>
> **A2**: Thank you for pointing this out. We agree that ODE-M incurs higher per-merge compute than algebraic baselines, since it requires ODE integration together with repeated first-order gradient evaluations. We therefore explicitly report the 20-task performance–efficiency trade-off below:
>
> |Method|ViT-B/32|ViT-B/16|ViT-L/14|
> |-|-:|-:|-:|
> |Task Arithmetic|21 s|29 s|42 s|
> |DARE|26 s|36 s|51 s|
> |TIES|31 s|43 s|61 s|
> |OPCM|58 s|81 s|112 s|
> |ODE-M|123.9 s|200.8 s|246.8 s|
>
> Although ODE-M is slower than OPCM, the overhead remains bounded in absolute terms: about 66s / 120s / 135s extra per merge on ViT-B/32 / ViT-B/16 / ViT-L/14, respectively. In return, ODE-M achieves consistently stronger final performance in the 20-task setting. For example, on ViT-L/14, ODE-M improves final ACC from 76.0 (OPCM) to 81.1; on ViT-B/32, it improves from 64.9 to 65.7; and on ViT-B/16, from 78.3 to 79.8. We therefore view this as a practical performance–efficiency trade-off rather than prohibitive overhead.
>
> In addition, this cost is incurred only at merge time, not during inference, and later merges under the standard schedule $\frac{1}{k}$ use a shorter effective integration horizon, so the runtime gap with OPCM gradually narrows as the stream grows. The current implementation is also not yet optimized for efficiency; the overhead can be further reduced through smaller calibration batches, fewer integration steps, mixed precision, and related engineering improvements. Overall, our results suggest that ODE-M introduces additional but still manageable compute overhead in exchange for consistently better continual merging performance.
>
> **Q3**: Sensitivity to the representativeness of the calibration set.
>
> **A3**: Empirically, the final performance does not appear to be overly sensitive to having a representative calibration set. In our main experiments(Table 1 & 2), the results are reported over 10 random seeds. Since the calibration set is randomly sampled in each run, the calibration data used in each run is in fact different. Despite this variation, ODE-M still achieves consistent performance, which suggests that the method does not rely on a uniquely representative calibration set.
>
> **Q4**: Potential combination of ODE-M with permutation-alignment techniques.
>
> **A4**: Thanks. We agree that the combination might be made more concrete. The two components are orthogonal in where they act in the pipeline: permutation alignment can be applied before each continual merge as an endpoint preprocessing step, in order to reduce neuron/channel mismatch between the current merged model and the incoming model. ODE-M can then be applied after alignment on the resulting aligned endpoints to construct a barrier-aware trajectory and determine the final operating point via rectification and time scheduling. In this sense, alignment addresses where the two endpoints are placed in parameter space, while ODE-M addresses how to move between them during continual merging. Therefore, the two are naturally composable, and a stronger alignment stage could plausibly further reduce the residual barrier that ODE-M needs to handle.
>
> We sincerely thank you again for your supportive and thoughtful review. We hope that the above clarifications have addressed your concerns and further clarified the design motivation, practical trade-offs, and possible extensions of ODE-M.

---

> > ### Author Rebuttal · Reviewer_YzKV · 2026-04-02
> >
> > The authors have addressed my concerns.

---

> > > ### Author Response · Authors · 2026-04-02
> > >
> > > Dear Reviewer YzKV,
> > >
> > > We sincerely appreciate your great efforts, constructive suggestions, and positive assessment you have provided once again! We are glad to hear that our responses have addressed your concerns. We are always willing to address any of your further concerns.
> > >
> > > Best Regards,
> > >
> > > Authors

---

### Official Review · Reviewer_x1p5 · 2026-03-12

**Soundness:** 2
**Presentation:** 2
**Significance:** 2
**Originality:** 3
**Overall Recommendation:** 3
**Confidence:** 4

**Summary:**

This work proposes a novel method for performing continual model merging using ODE between the model merged so far and the new model to merge. By trying to find the path that minimizes the barrier loss between the two models, they try to get to the best merged point that doesn't hurt the performance.

**Compliance With Llm Reviewing Policy:**

Affirmed.

**Final Justification:**

The authors addressed my concerns partially.

**Key Questions For Authors:**

1. In section 3.2, why the sign of <g_t,u_t> determines decreasing or increasing the loss?
2. In case 2 of page 13, why "if \Delta_L>0, then \gamma_t \leq \frac{\Delta_L}{<g_t,u_t>}" ?

**Limitations:**

Yes

**Strengths And Weaknesses:**

Strengths:
1. The proposed work is novel and has theoretical justification by relating the weight interpolation to the loss function and utilizing mode connectivity.
2. The proposed method has higher performance compared to previous static baselines.
3. They include heterogenous merging  scenarios where the importance of each task might be different.


Weaknesses:
1. The paper is not easy to read and needs better preparation or preliminary, especially for sections 3.1 and 3.2.
2. There is no comparison with model merging baselines [1-4] that use calibration data.
3. While the lack of experiments for language domain is discussed in the limitations, it would be interesting to see the performance for small language models like BERT or Roberta on text classification tasks or 1B models for generative tasks.
4. This paper [5] might be related or close to the proposed work and could be cited.

[1] AdaMerging: Adaptive Model Merging for Multi-Task Learning
[2] Activation-Informed Merging of Large Language Models
[3] Sens-Merging: Sensitivity-Guided Parameter Balancing for Merging Large Language Models
[4] Activation-Guided Consensus Merging for Large Language Models
[5] Multi-task learning via time-aware neural ODE

---

> ### Author Rebuttal · Authors · 2026-03-30
>
> Thank you for your insightful feedback. We appreciate your recognition of the novelty, motivation, and empirical strength of our work, and address your suggestions below:
>
> **Q1**: Clarity and presentation of Sec. 3.1 and 3.2.
>
> **A1**: We apologize for the lack of clarity in the current presentation. We will revise Sec. 3.1 and 3.2 to improve readability by adding more preliminaries, clarifying notation and definitions, and providing additional intuition and step-by-step explanations before the technical details. We will also improve the overall organization to make the paper easier to follow. If you would like more concrete details, we are eager to discuss these points in following discussion.
>
> **Q2**: Comparison with calibration-based model merging baselines.
>
> **A2**: Thank you for pointing us to these valuable activation- or calibration-based model merging works. We carefully examined them and found that [1], [3], and [4] are not originally designed for continual model merging (CMM). In our setting, CMM follows a strict sequential protocol: at step $k$, the algorithm only maintains the current merged model $\Psi_k$ and receives one new task model $\psi_{k+1}$, without retaining previous task-specific models or historical task vectors. Thus, a valid CMM method must work under this constraint.
>
> Specifically, [1] AdaMerging and [3] Sens-Merging both require explicit access to multiple task vectors/models simultaneously, and thus cannot be directly instantiated under CMM once past models have been absorbed into $\Psi_k$. [4] ACM, although proposed in an offline setting, can be adapted to CMM since its coefficients are computed relative to the shared pretrained model. In contrast, [2] AIM is a local activation-based module and can be directly applied to sequential updates. Therefore, we added new comparisons with the two applicable methods, [2] and [4]:
>
> |Methods|ViT-B/32 8 tasks|ViT-B/32 14 tasks|ViT-B/32 20 tasks|ViT-B/16 8 tasks|ViT-B/16 14 tasks|ViT-B/16 20 tasks|ViT-L/14 8 tasks|ViT-L/14 14 tasks|ViT-L/14 20 tasks|
> |-|-:|-:|-:|-:|-:|-:|-:|-:|-:|
> |AIM [2]|76.2|72.5|66.0|82.3|77.5|69.7|87.2|84.0|76.8|
> |ACM [4]|72.6|69.2|63.5|79.3|74.5|67.8|84.5|81.1|73.8|
> |OPCM|75.5|71.9|65.7|81.8|77.1|70.3|87.0|83.5|76.0|
> |ODE-M|79.6|74.1|67.1|85.3|78.5|70.4|87.7|85.8|81.1|
>
> As shown in Table, when compared against these activation-based baselines, ODE-M still achieves the best overall performance.
>
> **Q3**: Language-domain evaluation.
>
> **A3**: We agree that evaluating NLP tasks can provide a more comprehensive picture of the effectiveness of ODE-M. Due to the time constraint, we were only able to conduct additional experiments on Flan-T5-Base on 8 different NLP tasks:
>
> |Method|CoLA|MNLI|MRPC|QNLI|QQP|RTE|SST2|STSB|ACC|
> |-|-:|-:|-:|-:|-:|-:|-:|-:|-:|
> |FT|69.1|82.7|85.5|90.9|84.0|84.4|92.9|87.4|84.6|
> |SWA|69.1|62.6|79.4|89.8|83.9|81.2|91.7|73.2|78.8|
> |Task Arithmetic|70.5|57.8|78.4|90.2|83.6|80.5|92.3|77.8|78.9|
> |Ties-Merging|68.8|50.5|79.8|89.9|83.1|79.2|91.9|79.3|77.8|
> |OPCM|69.7|72.9|78.8|90.2|83.8|82.2|92.3|74.7|80.6|
> |ODE-M|70.8|75.6|80.5|90.5|83.8|82.5|92.5|78.9|81.9|
>
> The results still show clear gains of ODE-M over the merging baselines, achieving the best overall ACC, which further supports the effectiveness of our method in the language domain.
>
> **Q4**: Relevant prior work [5].
>
> **A4**: Thanks. We will cite [5] in the introduction and discuss its relation to our method more clearly.
>
> **Q5**: Clarification of why the sign of $\langle g_t,u_t\rangle$ determines local loss decrease or increase.
>
> **A5**: Thanks. The key point is that $\langle g_t,u_t\rangle$ is exactly the first-order directional derivative of the loss $L$ along the velocity direction $u_t$. By the chain rule, if $\theta(t)$ follows the dynamics $\dot{\theta}(t)=u_t$, then $\frac{\mathrm{d} }{\mathrm{d}t} L(\theta(t))=\langle \nabla_\theta L(\theta(t)),u_t\rangle=\langle g_t,u_t\rangle$. Hence, $\langle g_t,u_t\rangle$ is exactly the first-order directional derivative of the loss along $u_t$: a negative sign means the loss decreases locally, positive sign means it increases locally, and zero sign means no first-order change.
>
> **Q6**: Clarification of the condition $\gamma_t \leq \frac{\Delta_L}{<g_t,u_t>}$.
>
> **A6**: In case 2, we have $\langle g_t,u_t\rangle>0$, so $\frac{\Delta_L}{\langle g_t,u_t\rangle}$ is positive when $\Delta_L>0$. Since $\gamma_t=\text{clip}(\frac{\Delta_L}{\langle g_t,u_t\rangle},0,1)$, by the definition of clipping we always have $\gamma_t\leq\frac{\Delta_L}{\langle g_t,u_t\rangle}$.
>
> [1] AdaMerging: Adaptive Model Merging for Multi-Task Learning
> [2] Activation-Informed Merging of Large Language Models
> [3] Sens-Merging: Sensitivity-Guided Parameter Balancing for Merging Large Language Models
> [4] Activation-Guided Consensus Merging for Large Language Models
> [5] Multi-task learning via time-aware neural ODE
>
> Once again, we thank you for your thoughtful review and hope that the above clarifications address your concerns.

---

> > ### Author Rebuttal · Reviewer_x1p5 · 2026-04-04
> >
> > Thank you authors for your efforts in addressing my concerns. I'll need to look through the comments more carefully for a better assessment of the work. For now, I raise my score to 3.

---

> > > ### Author Response · Authors · 2026-04-06
> > >
> > > Dear Reviewer x1p5,
> > >
> > > Thank you for the time and effort you have devoted to reviewing our work. We sincerely appreciate your careful reading of our rebuttal and your willingness to raise your score.
> > >
> > > As the discussion period is now approaching its end, we would be very grateful for any further feedback you may have after looking through our responses more carefully. If there are still any points that you find unclear or any additional concerns you would like us to address, we are happy to respond as promptly and clearly as possible.
> > >
> > > Thank you again for your consideration and support.
> > >
> > > Best Regards,
> > >
> > > Authors

---

### Decision · Program_Chairs · 2026-04-30

**Decision:**

Accept (regular)

**Comment:**

This paper introduces a new perspective (ODE trajectories) to continual model merging by formulating the merging process as a continuous trajectory in parameter space governed by an ODE. The method aims to construct low-loss connecting paths to mitigate catastrophic forgetting. The primary weaknesses identified by the reviewers  include lack of NLP experiments, baseline comparisons, and missing ablations. During the rebuttal, the authors made significant efforts to address these concerns. They added a completely new suite of NLP experiments (Flan-T5 on 8 tasks), added new baselines (AIM, ACM), provided new runtime analyses, and added new empirical validations for their scheduling strategy.  Two reviewers were fully satisfied by the rebuttal. Reviewer S4Gp raised to 4 but remained concerned on efficiency. Reviewer x1p5 raised to 3 but left concerns partially unresolved due to limited engagement at the end of the discussion period. The core issue is that the amount of new material introduced during the rebuttal phase constitutes a major revision. The authors are strongly encouraged to incorporate all the new material generated during the rebuttal including i) including the NLP (Flan-T5) experiments to demonstrate cross-domain generalization; ii) including the runtime/compute trade-off tables to ensure transparency regarding the method's overhead; iii) integrating the clarifications regarding the practicality and size of the calibration set; and iv) revising Sections 3.1 and 3.2 for clarity, into the subsequent version.